# Robust State Estimation for Uncertain Discrete Linear Systems with Delayed Measurements

**Zhijun Li** [1,2]**, Minxing Sun** [1,2]**, Qianwen Duan** [1,2] **and Yao Mao** [1,2,*]

[1] Key Laboratory of Optical Engineering, Chinese Academy of Sciences, Chengdu 610209, China; zhijunhome_ioe@163.com (Z.L.); sunminxing20@mails.ucas.ac.cn (M.S.); duanqianwen16@mails.ucas.ac.cn (Q.D.)

[2] Institute of Optics and Electronics, Chinese Academy of Sciences, Chengdu 610209, China

[*] Correspondence: maoyao@ioe.ac.cn

**Abstract:** Measurement delays and model parametric uncertainties are meaningful issues in actual systems. Addressing the simultaneous existence of random model parametric uncertainties and constant measurement delay in the discrete-time linear systems, this study proposes a novel robust estimation method based on the combination of Kalman filter regularized least-squares (RLS) framework and state augmentation. The state augmentation method is elaborately designed, and the cost function is improved by considering the influence of modelling errors. A recursive program similar to the Kalman filter is derived. Meanwhile, the asymptotic stability conditions of the proposed estimator and the boundedness conditions of its error covariance are analyzed theoretically. Numerical simulation results show that the proposed method has a better processing capability for measurement delay and better robustness to model parametric uncertainties than the Kalman filter based on nominal parameters.

**Keywords:** constant measurement delay; random parametric uncertainties; state augmentation; robust state estimation; regularized least squares

**MSC:** 49



## 1. Introduction

State estimation, which is generally applied in automatic control and signal processing, is a method for estimating the internal state of a dynamic system based on available measurement data. For linear systems with external disturbances of normal distribution characteristics, the standard Kalman filter (SKF) is the optimal filter under the minimum mean square error (MMSE) criterion and is extensively used in many fields [1–4] such as control, finance, communication, etc. The traditional state estimation methods assume that the measured data is transmitted to the filter with no delay. However, in practical systems, such as spacecraft systems [5], satellite [6], and photoelectric tracking systems [7], all the measurements have time delays, which are mainly composed of the acquisition time, processing time, and transmission time of the sensor data. Meanwhile, the modelling errors are generally unavoidable [8], and will further influence the performance of the systems. Therefore, both measurement delay and random model parametric uncertainties are the interest of this article.

Plenty of detectors have the characteristic of constant measurement delay. Frequently occurring severe network congestion or packet loss may cause a time-varying delay in many systems [9–11]. Most sensors have similar or slightly changed measurement delay because of their fixed transmission environment, acquisition environment, and acquisition algorithm. Considering these subtle changes would have a weaker influence in the discrete domain, the measurement delay can be reasonably treated as a constant. For instance, in the field of our research, the target detector in the photoelectric tracking system has a

considerable constant measurement delay, which is an increasingly attractive issue in high-precision tracking. Many innovative methods [7,12,13] have been proposed to compensate its affect on tracking performance.

The measurement delay problem in state estimation is called the out-of-sequence measurement (OOSM) problem, the time-delayed measurement problem, or the time-varying measurement problem [14]. For the problems with known delays, backward prediction, state augmentation, and extrapolation are three distinctive approaches. The backward prediction (or retrodiction) was originally proposed by Bar-Shalom [15] to solve the one-step delayed OOSM problem under the Kalman filter algorithm framework. In this approach, the state and covariance matrices at the time OOSM occurs are backward predicted when the filtering system receives the OOSM. Then, the delayed measurement is utilized to update the current state and covariance. When the process noise satisfies the discrete continuous-time model, the approach can achieve optimal performance.The extended version for multiple-step delayed OOSM is proposed in [16]. Zhang et al. [17] proposed a sub-optimal version to reduce the computing burden under certain circumstances. The second valuable approach for the time-delayed measurement problem is summarized as the state augmentation approach, in which the delayed measurement is used to estimate the state of the corresponding past moment, and the prediction of the current state is obtained from the corrected past state. The key point to this approach is to elaborately augment the state vector and establish the correlation between the augmented state vector containing the corresponding past state and the delay measurement. One estimator based on the augmented state vector is proposed in [18] to deal with the problem that the change of current state could be affected by the d-step preceding state. However, it could not be adopted to solve the problem in this paper that detectors have measurement delay and the preceding state has no influence in the target's state transition. Based on the Bayesian theory, a state augmentation Kalman filter solution to the time-delayed measurement problem is suggested in [19]. The third practical approach is called the extrapolation approach. In [19], by assuming that the current measurement residual is equal to the residual at the corresponding time of the OOSM, the measurement for the current time is calculated by extrapolation. Then, the current state is estimated by incorporating the extrapolated measurement into the Kalman filter. Ref. [7] suggested to use the delayed measurement to estimate the state at the corresponding time first, then use the process matrix to iteratively multiply the past state to extrapolate the current state.

The researches on the time-delayed measurement problem are fruitful. However, the problem that this article focuses on is the simultaneous existence of constant measurement delay and random model parametric uncertainties in state estimation.

The estimator to solve the problem of random parametric uncertainties in state estimation is collectively referred to as a robust filter/estimator. Here are three representative robust state estimation methods. The H-infinity filter is developed based on the H-infinity linear control theory proposed by Zames in 1980 [20,21]. Unlike the Kalman filter that uses the mean square error criterion, the H-infinity filter adopts the minimax criterion to minimize the maximum estimation error. It does not need to know prior information such as the statistical characteristics of environmental noise, and can minimize the influence of external interference on the state estimation results. Therefore, this method is more robust to system model errors and external interference. The blemish of the H-infinity filter is that it needs to continuously test specific existence conditions when performing recursive filtering operations. If these conditions fail during any iteration, the desired performance will be lost and the filter may diverge. The method of set-valued estimation assumes that the noise disturbance of the measurement is norm-bounded. Based on this assumption, ellipsoids are constructed around the state estimate [22,23]. Here again, this method is encountered with the requirement of inspecting for certain existence conditions, which may limit the application of this method to recursive filtering. The third solution to the robust filter design is a regularized least-squares (RLS)-based framework, which is first proposed by Sayed in [24]. In this framework, the standard Kalman filter is regarded as the solution

of a regularized least squares problem. Then, the objective function of this regularized least-squares problem is further improved considering the uncertainties of model parameters. Although this robust filter focuses on a worst-case analysis that may be conservative under relatively "small" uncertainties, it has many attractive properties. For example, it does not need to verify certain existence conditions at every moment of recursive filtering; it has a similar structure to the Kalman filter, and it also has low computational complexity, etc. Therefore, scholars have further extended this robust filter framework. In [25], a new robust filter method that weighs nominal performance and uncertainty is proposed. Refs. [8,26] propose sensitivity penalization-based robust state estimation methods.

According to the above analysis, the research studies on the model parametric uncertainties are also prolific. However, in the public references, few papers specifically address the method of simultaneously overcoming the constant measurement delay and random model parametric uncertainties in the discrete-time linear systems. Fortunately, there are many similar studies in the published literature for reference. For example, the robust estimation problem of a class of discrete-time systems with delays and lossy measurements is studied in [27]. Compared with the problem studied in this paper, the robust H-infinity filter designed by [27] focuses more on the reduction of delay-dependent conditions and the processing of lossy measurements. Basically, there have been two ways to model the measurement of missing phenomena, that is, using a binary switching sequence and using a Markov chain. Refs. [27–29] all pay more attention to the problem of lossy measurements. Excessive attention leads to these similar works, unable to focus on solving the problems of the content of this research. Meanwhile, in [27], the H-infinity filtering method is used to handle the model parametric uncertainties. As mentioned earlier, the H-infinity filter needs to continuously test specific existence conditions when performing recursive filtering operations. To solve the problem of model parametric uncertainties, in [27], the robust Kalman filtering is derived in the linear minimum variance sense by using the innovation analysis approach. The dimension of the designed filter is the same as the original systems. However, the recursive filtering process is complicated and is not conducive to engineering realization. In [27], the robust recursive estimator is designed based on the orthogonal projection theorem. The stochastic uncertainties of the system model are described by multiplicative noises, which lead to a narrow applicability of the estimator to the actual systems. For more specific issues, the attitude estimation filtering problem with model uncertainties in the state, output, and process noise matrices and star sensor delays has been studied in [30]. In [30], the uncertain attitude estimation model is established for the actual system. Combined with star sensor delays, a new finite-horizon robust Kalman filter design is derived for the uncertain attitude estimation system. The optimized filter parameters can be obtained to minimize the upper bound on the estimation error covariance. However, the method proposed in [30] is not universal enough, its application scenario is limited to a specific type of system, and the assumption about measurement delay is also different from this research. In addition, for the model uncertainties and time-delayed measurements of different types of nonlinear systems, refs. [31–33] are worthy of reference. According to the above analysis, similar published documents are not suitable for solving the problems of this research. This research should provide a more focused and universal estimator design solution, and the final-designed estimator should be simple, reliable, and easy to implement.

Motivated by the aforementioned analysis, this study suggests a novel robust estimation method, which combines an RLS-based robust filter design framework and state augmentation. The main contributions of the paper are as follows: (1) This study elaborately designs the specific state augmentation method to deal with the constant measurement delay, and modifies the cost function of the Kalman filter RLS problem for the random model parametric uncertainties. (2) Based on this design, a recursive filtering procedure is derived. As long as the probability distribution of parametric uncertainties are known and the two matrices related to the filter are calculated offline in advance, online filtering can be performed. Compared with the similar works in [27–29], the recursive filtering

procedure designed in this study has similar computational complexity to the Kalman filter. Meanwhile, it does not need to design optimal parameters and continuously test specific existence conditions, which shows that the proposed estimator is simple and easy to implement. (3) The asymptotic stability conditions and the error covariance boundedness conditions of the proposed estimator are derived to guarantee the reliability of the proposed estimator. (4) Besides, this paper designs numerical simulations to verify the effectiveness of the proposed estimator.

The remaining structural arrangement of this article is as follows: The problem statement and the design of robust state estimation are presented in Section 2. Section 3 focuses on the recursive procedure, the asymptotic stability conditions of the estimator as well as the conditions for the boundedness of the estimation error matrix. The numerical simulations and verification of practical experimental system are discussed in Section 4. Conclusions are presented in Section 5. The appendices exhibit the derivation of recursive estimation procedures and proof of the proposed theory in this study.

Notations: Suppose $x$ is a column vector and $W$ is a positive definite matrix. Define $\|x\|$ and $\|x\|_W$ to represent the Euclidean norm of $x$ and its weighted form, respectively. That is, $\|x\| = \sqrt{x^T x}$ and $\|x\|_W = \sqrt{x^T W x}$. $\delta_{kj}$ is the Kronecker delta function and $col\{X_j\}$ represents the vector/matrix stacked by $X_j$. $E(*)$ expresses the mathematical expectation of a stochastic variable, vector, or matrix. $\mathbb{R}^n$ represents an n-dimensional Euclidean space. $0_{m \times n}$ represents a matrix of all zeros with $n$ rows and $m$ columns. **diag**$\{A, B\}$ is a simplified representation of $\begin{bmatrix} A & 0 \\ 0 & B \end{bmatrix}$.

## 2. Problem Statement and the Design of Robust State Estimator

Consider the following discrete-time linear system with constant measurement delay and random parametric uncertainties:

$$\begin{cases} x_{k+1} = A_k(\varepsilon_k)x_k + B_k(\varepsilon_k)w_k \\ \quad y_k = C_{k-d}(\varepsilon_{k-d})x_{k-d} + v_k \end{cases} \tag{1}$$

where $x_k \in \mathbb{R}^n$ is the state vector, $y_k \in \mathbb{R}^m$ is the measurement vector, and $d$ is the frames of the measurement delay. The modelling error $\varepsilon_k$ is composed of $L$ real-valued bounded scalars $\varepsilon_{i,k}, i = 1, \cdots, L$. $\varepsilon_{i,k}$ represents the parametric modelling error at moment $k$ in the $i$-th experiment. That is $E = \{\varepsilon_k | |\varepsilon_{k,i}| \leq 1, i = 1, \cdots, L\}$. $A_k(\varepsilon_k), B_k(\varepsilon_k), C_k(\varepsilon_k)$ are matrices related to $\varepsilon_k$ and of size $n \times n, n \times n, m \times n$. $w_k \in \mathbb{R}^n$ and $v_k \in \mathbb{R}^m$ are uncorrelated and gaussian random noise with variances $Q$ and $R$, respectively. That is,

$$\begin{cases} E\left[w_k w_j^T\right] = \delta_{kj}Q \geq 0 \\ E\left[v_k v_j^T\right] = \delta_{kj}R \geq 0 \, . \\ E\left[w_k v_j^T\right] = 0 \end{cases} \tag{2}$$

**Remark 1.** *Unlike [8,24,34], System (1) neither requires the system matrices to be linearly dependent on the norm bounded uncertainty matrix nor requires the elements of the system matrices to be differentiable functions of $\varepsilon_k$. The way the modelling errors $\varepsilon_k$ affect the plant parameters can be arbitrary. This feature makes it possible for the System (1) to capture more actual-system behavior than the ones in [8,24,34].*

Assuming that $d$ is known and time-invariant, System (1) can be equivalently reconstructed into a delay-free System (3).

$$\begin{cases} \bar{x}_{k+1} = \bar{A}_k(\varepsilon_k)\bar{x}_k + \bar{B}_k(\varepsilon_k)w_k \\ \quad y_k = \bar{C}_k(\varepsilon_k)\bar{x}_k + v_k \end{cases} \tag{3}$$

in which, $\bar{x}_k = \left[x_k^T, \Delta_k^T, \Delta_{k-1}^T, \ldots, \Delta_{k-d+1}^T\right]^T$, $\Delta_k = C_{k-1}(\varepsilon_{k-1})x_{k-1} - C_k(\varepsilon_k)x_k$. Furthermore, the definitions of $\bar{A}_k(\varepsilon_k), \bar{B}_k(\varepsilon_k)$, and $\bar{C}_k(\varepsilon_k)$ are shown in (4).

$$
\begin{aligned}
\bar{A}_k(\varepsilon_k) &= \begin{bmatrix} A_k(\varepsilon_k) & 0_{n\times(d-1)m} & 0_{n\times m} \\ C_k(\varepsilon_k) - C_{k+1}(\varepsilon_{k+1})A_k(\varepsilon_k) & 0_{m\times(d-1)m} & 0_{m\times m} \\ 0_{(d-1)m\times n} & I_{(d-1)m} & 0_{(d-1)m\times m} \end{bmatrix} \\
\bar{B}_k(\varepsilon_k) &= \begin{bmatrix} B_k(\varepsilon_k) \\ -C_{k+1}(\varepsilon_{k+1})B_k(\varepsilon_k) \\ 0_{(d-1)m\times n} \end{bmatrix}, \bar{C}_k(\varepsilon_k) = \begin{bmatrix} C_k(\varepsilon_k), \overbrace{I_m, \cdots, I_m}^{d} \end{bmatrix}
\end{aligned}
\tag{4}
$$

After converting System (1) to an augmented delay-free model, the random parametric uncertainties of the system are considered. According to [24,35], a clear explanation of the SKF is to solve the RLS (regularized least squares) problem which is presented in (5).

$$
\begin{pmatrix} \hat{x}_{k|k+1} \\ \hat{w}_{k|k+1} \end{pmatrix} = \arg\min_{x_k, w_k} \left[ \left\| x_k - \hat{x}_{k|k} \right\|_{P_{k|k}^{-1}}^2 + \|w_k\|_{Q_k^{-1}}^2 + \|y_{k+1} - Cx_{k+1}\|_{R_{k+1}^{-1}}^2 \right] .
\tag{5}
$$

Considering the estimation performance deterioration caused by modeling errors, the cost function of the RLS problem is expanded as follows:

$$
\begin{aligned}
J(\alpha_k) &= E\left\{ \left\| \bar{x}_k - \hat{\bar{x}}_{k|k} \right\|_{P_{k|k}^{-1}}^2 + \|w_k\|_{Q_k^{-1}}^2 + \left\| y_{k+1} - \bar{C}_{k+1}(\varepsilon_{k+1})\bar{x}_{k+1} \right\|_{R_{k+1}^{-1}}^2 \right\} \\
&= \|\alpha_k\|_{\Phi_k}^2 + E\left\{ \|H_k(\varepsilon_k, \varepsilon_{k+1})\alpha_k - \beta_k(\varepsilon_k, \varepsilon_{k+1})\|_{\Psi_k}^2 \right\}.
\end{aligned}
\tag{6}
$$

where,

$$
\begin{cases}
\Psi_k = R_{k+1}^{-1} \\
H_k(\varepsilon_k, \varepsilon_{k+1}) = \bar{C}_k(\varepsilon_{k+1})\begin{bmatrix} \bar{A}_k(\varepsilon_k) & \bar{B}_k(\varepsilon_k) \end{bmatrix} \\
\beta_k(\varepsilon_k, \varepsilon_{k+1}) = y_{k+1} - \bar{C}_k(\varepsilon_{k+1})\bar{A}_k(\varepsilon_k)\hat{\bar{x}}_{k|k} \\
\Phi_k = \mathbf{diag}\left\{ P_{k|k}^{-1}, Q_k^{-1} \right\} \\
\alpha_k = col\left\{ \bar{x}_k - \hat{\bar{x}}_{k|k}, w_k \right\}
\end{cases}
\tag{7}
$$

In (6), the expanded cost function has used mathematical expectations to handle the random parametric uncertainties of the augmented delay-free model. When there is no modelling error, the state estimator in (6) degenerates into a SKF. According to the definitions of $\Phi_k$ and $\Psi_k$, the cost function $J(\alpha_k)$ is a strictly convex function. That is, there is a global minimum $\alpha_{kopt}$ at $\partial J(\alpha_k)/\partial\alpha_k = 0$. Expanding the cost function in (6),

$$
\begin{aligned}
J(\alpha_k) = \alpha_k^T \Phi_k \alpha_k &+ E\left\{ \alpha_k^T H_k(\varepsilon_k, \varepsilon_{k+1})^T \Psi_k H_k(\varepsilon_k, \varepsilon_{k+1})\alpha_k \right. \\
&+ \beta(\varepsilon_k, \varepsilon_{k+1})^T \Psi_k \beta(\varepsilon_k, \varepsilon_{k+1}) - \alpha_k^T H_k(\varepsilon_k, \varepsilon_{k+1})^T \Psi_k \beta(\varepsilon_k, \varepsilon_{k+1}) \\
&\left. - \beta(\varepsilon_k, \varepsilon_{k+1})^T \Psi_k H_k(\varepsilon_k, \varepsilon_{k+1})\alpha_k \right\}
\end{aligned}
\tag{8}
$$

is obtained. Find the partial derivative of (8) for $\alpha_k$ and let $\partial J(\alpha_k)/\partial\alpha_k = 0$,

$$
\begin{aligned}
&\left( \Phi_k + E\left\{ H_k(\varepsilon_k, \varepsilon_{k+1})^T \Psi_k H_k(\varepsilon_k, \varepsilon_{k+1}) \right\} \right)\alpha_k \\
&= E\left\{ H_k(\varepsilon_k, \varepsilon_{k+1})^T \Psi_k \beta(\varepsilon_k, \varepsilon_{k+1}) \right\}
\end{aligned}
\tag{9}
$$

is obtained. Substituting (7) into (9), (10) is obtained.

$$
\begin{aligned}
\left(\begin{bmatrix} P_{k|k}^{-1} & 0 \\ 0 & Q_k^{-1} \end{bmatrix} + E\left\{ \begin{bmatrix} \bar{A}_k^T(\varepsilon_k) \\ \bar{B}_k^T(\varepsilon_k) \end{bmatrix} \bar{C}_{k+1}^T(\varepsilon_{k+1}) \right. \right. \\
\left. \left. \times R_{k+1}^{-1} \bar{C}_{k+1}(\varepsilon_{k+1}) \begin{bmatrix} \bar{A}_k(\varepsilon_k) & \bar{B}_k(\varepsilon_k) \end{bmatrix} \right\} \right) \begin{bmatrix} \bar{x}_k - \hat{\hat{x}}_{k|k} \\ w_k \end{bmatrix} \\
= E\left\{ \begin{bmatrix} \bar{A}_k^T(\varepsilon_k) \\ \bar{B}_k^T(\varepsilon_k) \end{bmatrix} \bar{C}_{k+1}^T(\varepsilon_{k+1}) \right\} R_{k+1}^{-1} y_{k+1} \\
- E\left\{ \begin{bmatrix} \bar{A}_k^T(\varepsilon_k) \\ \bar{B}_k^T(\varepsilon_k) \end{bmatrix} \bar{C}_{k+1}^T(\varepsilon_{k+1}) R_{k+1}^{-1} \bar{C}_{k+1}(\varepsilon_{k+1}) \bar{A}_k(\varepsilon_k) \right\} \hat{\hat{x}}_{k|k}
\end{aligned}
\tag{10}
$$

The following matrices are defined for further simplification.

$$
\begin{cases}
H_{k1} = E\left\{ \begin{bmatrix} \bar{A}_k^T(\varepsilon_k) \\ \bar{B}_k^T(\varepsilon_k) \end{bmatrix} \bar{C}_{k+1}^T(\varepsilon_{k+1}) R_{k+1}^{-1} \bar{C}_{k+1}(\varepsilon_{k+1}) \begin{bmatrix} \bar{A}_k(\varepsilon_k) & \bar{B}_k(\varepsilon_k) \end{bmatrix} \right\} \\
H_{k2} = \begin{bmatrix} \bar{A}_k^T(\varepsilon_k) \\ \bar{B}_k^T(\varepsilon_k) \end{bmatrix} \bar{C}_{k+1}^T(\varepsilon_{k+1}) \\
H_{k3} = E\left\{ \begin{bmatrix} \bar{A}_k^T(\varepsilon_k) \\ \bar{B}_k^T(\varepsilon_k) \end{bmatrix} \bar{C}_{k+1}^T(\varepsilon_{k+1}) R_{k+1}^{-1} \bar{C}_{k+1}(\varepsilon_{k+1}) \bar{A}_k(\varepsilon_k) \right\}
\end{cases}
\tag{11}
$$

Obviously, $H_{k3} = H_{k1} \begin{bmatrix} I_{(n+dm)} & 0_{(n+dm)\times n} \end{bmatrix}'$. Finally, (10) can be simplified as

$$
(\Phi_k + H_{k1})\alpha_k = H_{k2} R_{k+1}^{-1} y_{k+1} - H_{k3}\hat{\hat{x}}_{k|k}.
\tag{12}
$$

**Remark 2.** *Assume that the statistical property of $\varepsilon_k$ is known, if the relationships between the modelling errors $\varepsilon_k$ and matrices A, B, and C are simple, $H_{k1}$ and $H_{k2}$ can be solved by direct algebraic operations. Otherwise, $H_{k1}$ and $H_{k2}$ are calculated by stochastic simulations [36]. For example, according to the statistical property of $\varepsilon_k$, 10,000 realizations of $H_{k1}$ and $H_{k2}$ by stochastic simulations are constructed, then the $H_{k1}$ and $H_{k2}$ can be calculated by averaging. In contrast, if the statistical property of $\varepsilon_k$ is not known, $H_{k1}$ and $H_{k2}$ can also be calculated according to plenty of realizations of matrices A, B, and C obtained during the modelling process. According to the above analysis, it can be obtained that $H_{k1}$ and $H_{k2}$ can be calculated offline, which means that the filter proposed in this study may be more conducive to implementation and application than those in [8,24,34].*

**Remark 3.** *The proposed estimator is characterized by the ingenious combination of the Kalman filter RLS framework and state augmentation, which can cope with the simultaneous existence of model parametric uncertainties and time-delayed measurement. Since the proposed estimator is an improvement on the Kalman filter RLS framework, the proposed estimator has a recursive filtering form similar to the Kalman filter, which is conducive to engineering realization. Moreover, the proposed estimator does not require additional parameter design and optimization, as well as the online test of certain existing conditions. This property makes the proposed estimator more reliable and convenient. However, the computational cost of the state augmentation in the proposed estimator cannot be ignored. Fortunately, this article is not aimed at long-term trajectory prediction, and a short-term measurement delay that can be regarded as constant in most systems. For example, the time delay of the target detector of the photoelectric tracking system is generally 2 to 4 frames [7], and the computational cost of state augmentation is affordable.*

## 3. Recursive Procedure and Asymptotic Stability Conditions of the Estimator

Based on the analysis and explanation in Section 2, the recursive procedure is provided in this section. Simultaneously, the asymptotic stability conditions of the proposed estimator and the conditions for the boundedness of estimation error matrix are explicitly presented. The recursive procedure is composed of three steps. The first step is initialization.

$$\hat{x}_{0|0} = P_{0|0}E\{\bar{C}_0(\varepsilon_0)\}R_0^{-1}y_0, \, P_{0|0} = \left(\bar{\Pi}_0^{-1} + E\{\bar{C}_0^T(\varepsilon_0)R_0^{-1}\bar{C}_0(\varepsilon_0)\}\right)^{-1} \tag{13}$$

In (13), $\bar{\Pi}_0 = E\left\{(\bar{x}_0 - E(\bar{x}_0))(\bar{x}_0 - E(\bar{x}_0))^T\right\}$. The second step is parameter modification. A matrix is defined as

$$\begin{aligned} G_k &\overset{\Delta}{=} H_{k1} - \left[\begin{array}{c} \bar{A}_k^T(0) \\ \bar{B}_k^T(0) \end{array}\right]\bar{C}_{k+1}^T(0)R_{k+1}^{-1}\bar{C}_{k+1}(0)\left[\begin{array}{cc} \bar{A}_k(0) & \bar{B}_k(0) \end{array}\right] \\ &= \left[\begin{array}{cc} G_{k11} & G_{k12} \\ G_{k12}^T & G_{k22}^T \end{array}\right] \end{aligned} \tag{14}$$

in which, $G_k$ is a $(2n+md) \times (2n+md)$ matrix, $G_{k11}$, $G_{k12}$, and $G_{k22}$ are $(n+md) \times (n+md)$, $(n+md) \times n$, and $n \times n$ matrices, respectively. The definitions of the matrices $\hat{A}_k(0)$, $\hat{B}_k(0)$, $\hat{P}_{k|k}$ and $U_k$ are

$$\begin{cases} \hat{P}_{k|k} = \left(P_{k|k}^{-1} + G_{k11}\right)^{-1} \\ U_k = \left(Q_k^{-1} + G_{k22} - G_{k12}^T\hat{P}_{k|k}G_{k12}\right)^{-1} \\ \hat{B}_k(0) = \bar{B}_k(0) - \bar{A}_k(0)\hat{P}_{k|k}G_{k12} \\ \hat{A}_k(0) = \left(\bar{A}_k(0) - \hat{B}_k(0)U_kG_{k12}^T\right)\left(I_{n+md} - \hat{P}_{k|k}G_{k11}\right) \end{cases} \tag{15}$$

In the third step, the state estimation $\hat{x}_{k+1|k+1}$ is calculated by updating $P_{k+1|k}$, $R_{e,k+1}$, $P_{k+1|k+1}$. The definitions of updated formulas $P_{k+1|k}$, $R_{e,k+1}$, $P_{k+1|k+1}$ are as follows:

$$\begin{cases} P_{k+1|k} = \bar{A}_k(0)\hat{P}_{k|k}\bar{A}_k^T(0) + \hat{B}_k(0)U_k\hat{B}_k^T(0) \\ R_{e,k+1} = R_{k+1} + \bar{C}_{k+1}(0)P_{k+1|k}\bar{C}_{k+1}^T(0) \\ P_{k+1|k+1} = P_{k+1|k} - P_{k+1|k}\bar{C}_{k+1}^T(0)R_{e,k+1}^{-1}\bar{C}_{k+1}(0)P_{k+1|k} \end{cases} \tag{16}$$

Then

$$\begin{aligned} \hat{x}_{k+1|k+1} = {}& \hat{A}_k(0)\hat{x}_{k|k} + P_{k+1|k+1} \\ &\times \left(P_{k+1|k}^{-1}\left(\bar{A}_k(0)\hat{P}_{k|k}\left[\begin{array}{cc} I_{n+md} & 0 \\ & {}_{(n+md)\times n} \end{array}\right]\right.\right. \\ &\left.+ \hat{B}_k(0)U_k\hat{B}_k^T(0)\left[\begin{array}{cc} -G_{k12}^T\hat{P}_{k|k} & I_n \end{array}\right]\right)H_{k2}R_{k+1}^{-1}y_{k+1} \\ &\left.- \bar{C}_{k+1}^T(0)R_{k+1}^{-1}\bar{C}_{k+1}(0)\hat{A}_k(0)\hat{x}_{k|k}\right) \end{aligned} \tag{17}$$

The specific derivations of the recursive procedure is provided in Appendix A.

The asymptotic stability of the proposed estimator is discussed next. Suppose that the modelling errors $\varepsilon_{k,i}$ are normalized to be contracted, a set $E$ can be constituted as $E = \{\varepsilon||\varepsilon_{k,i}| \le 1, i = 1, \cdots, L\}$. For the convenience of discussion, the matrices $\bar{A}_k(0)$, $\bar{B}_k(0)$, $\bar{C}_k(0)$ are abbreviated as $\bar{A}_k$, $\bar{B}_k$, $\bar{C}_k$. Then, related matrices are defined as follows:

$$\begin{aligned} U_k &= \left(Q_k^{-1} + G_{k22} - G_{k12}^TG_{k11}^{-1}G_{k12}\right)^{-1}, \, D_k = \bar{B}_kU_k^{1/2} \\ J_k &= \left[\begin{array}{cc} 0 & U_k^{1/2}G_{k12}^TG_{k11}^{-1/2} \end{array}\right] \\ W_k &= \left[\begin{array}{cc} I & 0 \\ 0 & I + G_{k11}^{-1/2}G_{k12}U_kG_{k12}^TG_{k11}^{-1/2} \end{array}\right], \, F_k = \left[\begin{array}{c} R_k^{-1/2}\bar{C}_k \\ G_{k11}^{1/2} \end{array}\right] \end{aligned} \tag{18}$$

**Theorem 1.** *Assuming that $\bar{A}_k, \bar{B}_k, \bar{C}_k, R_k, Q_k, H_{k1}, H_{k2}$ are time-invariant, $(A, C)$ is detectable, $rank(A) = n$, and $(M1, M2)$ is stabilizable, the estimator is asymptotically stable in this study. The definitions of M1 and M2 are as follows:*

$$M1 = \begin{bmatrix} A & 0_{n \times (d-1)m} & 0_{n \times m} \\ C - CA & 0_{m \times (d-1)m} & 0_{m \times m} \\ 0_{(d-1)m \times n} & I_{(d-1)m \times (d-1)m} & 0_{(d-1)m \times m} \end{bmatrix} - \begin{bmatrix} B \\ -CB \\ 0_{(d-1)m \times n} \end{bmatrix} \Omega 1, M2 = \begin{bmatrix} B \\ -CB \\ 0_{(d-1)m \times n} \end{bmatrix} \Omega 2$$

where

$$\Omega 1 = \left( Q_k^{-1} + G_{k22} - G_{k12}^T G_{k11}^{-1} G_{k12} \right)^{-1} G_{k12}^T$$

$$\times \left( I + G_{k11}^{-1} G_{k12} \left( Q_k^{-1} + G_{k22} - G_{k12}^T G_{k11}^{-1} G_{k12} \right)^{-1} G_{k12}^T \right)^{-1}$$

$$\Omega 2 = \left( Q_k^{-1} + G_{k22} - G_{k12}^T G_{k11}^{-1} G_{k12} \right)^{-1/2}$$

$$\times \left( I + \left( Q_k^{-1} + G_{k22} - G_{k12}^T G_{k11}^{-1} G_{k12} \right)^{-1/2} G_{k12}^T G_{k11}^{-1} G_{k12} \right.$$

$$\times \left. \left( Q_k^{-1} + G_{k22} - G_{k12}^T G_{k11}^{-1} G_{k12} \right)^{-1/2} \right)^{-1/2}.$$

The proof of Theorem 1 is postponed to Appendix B.

**Theorem 2.** *Assuming that System (3) is exponentially stable in the sense of Lyapunov and the relevant matrices $\bar{A}_k, \bar{B}_k, \bar{C}_k, R_k, Q_k$ are all bounded for $k > 0$ and $\varepsilon_k \in E$, all conditions and assumptions of Theorem 2 are satisfied. Then, the estimation error covariance matrix of the proposed estimator is bounded.*

The proof of Theorem 2 is postponed to Appendix C.

## 4. Numerical Simulations

Before verifying the overall scheme of the proposed estimator, the processing capability of the proposed state augmentation method on the time-delayed measurements is verified first. Without loss of generality, a single-axis constant velocity model is employed to simulate the movement of a target. The state vector of the system is composed of the target's position and velocity. (19) specifies the detailed system model parameters.

$$A = \begin{bmatrix} 1 & T \\ 0 & 1 \end{bmatrix}, B = \begin{bmatrix} 1 & 0 \\ 0 & 1 \end{bmatrix}, C = \begin{bmatrix} 1 & 0 \end{bmatrix}, Q = \begin{bmatrix} 1.9 & 0 \\ 0 & 0.5 \end{bmatrix},$$
$$R = 1, x_0 = \begin{bmatrix} -30.04 \\ -3.492 \end{bmatrix}, \Pi_0 = \begin{bmatrix} 1 & 0 \\ 0 & 1 \end{bmatrix}. \tag{19}$$

In (19), $T = 0.01$ represents the sampling period, $Q$ and $R$ represent the process noise covariance and measurement noise covariance, respectively. $x_0$ is the initial state vector. $\Pi_0$ is the initial estimation error covariance.

Figure 1 shows the simulated target trajectory and its measurements with Gaussian noise and three-frame time delay. These measurements are used as the input of the standard KF and the state augmentation-based KF. Figure 2 shows the estimation errors of the two methods. According to the comparison results in Figure 2, it can be seen that the proposed state augmentation method effectively reduces the estimation error in the measurement delay situation because the proposed state augmentation method indirectly establishes the correlation between the delayed measurement and the current state.

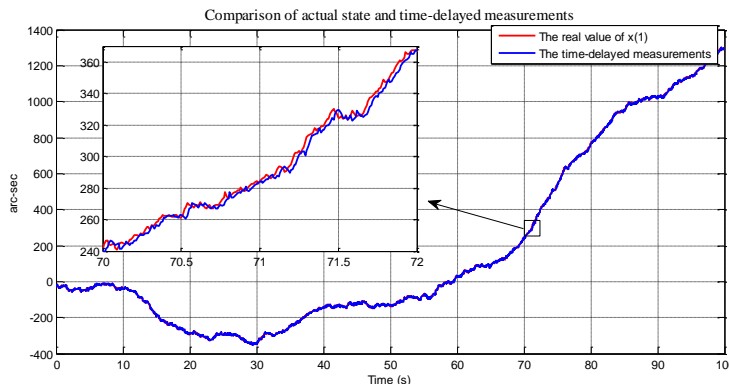

**Figure 1.** Actual state and time-delayed measurements.

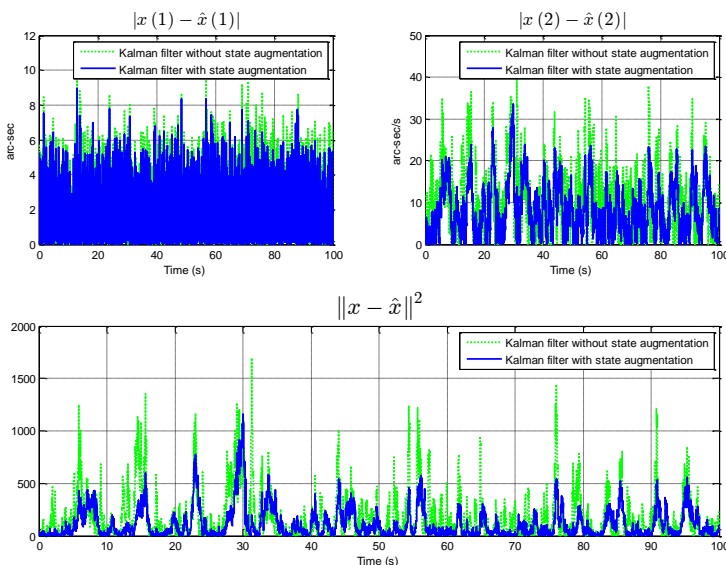

**Figure 2.** Comparison of estimation errors between SKF and improved KF using state augmentation.

In the proposed method, by elaborately designing a state augmentation method, the relationship between the delayed measurements and the current state is established, and then the original model is converted into a delay-free augmentation model. Since the current state is included in the augmented state vector, the augmented delay-free model can be further used in the filter design. If the system has no model parametric uncertainty, then the standard Kalman filter is the optimal linear estimator. When considering the influence of random parametric uncertainties, this work enters the mathematical expectation method to improve the cost function of the RLS-based Kalman filter framework. The recursive filtering procure derived from the modified cost function has a significant feature; that is, when the system model does not have random parametric uncertainties, the estimator degenerates into standard Kalman filter. When the system has random parametric uncertainties, the estimator can effectively suppress the influence of random parametric uncertainties on the estimation performance. In order to verify this part of the work individually, this research further designs a simulation experiment.

Without loss of generality, we add uncertain parameters with known statistical characteristics to the single-axis constant velocity model, and then compare the estimation performance between the standard Kalman filter and the proposed estimator to illustrate the effectiveness of this part of the work. The specific model parameters are as follows:

$$A = \begin{bmatrix} 1 & 0.01 + 0.005 \cdot \zeta \\ 0 & 1 \end{bmatrix}, B = \begin{bmatrix} 1 & 0 \\ 0 & 1 \end{bmatrix}, C = \begin{bmatrix} 1 & 0 \end{bmatrix},$$
$$Q = \begin{bmatrix} 1.9 & 0 \\ 0 & 0.5 \end{bmatrix}, R = 1, x_0 = \begin{bmatrix} -35.7101 \\ -10.4338 \end{bmatrix}, \Pi_0 = \begin{bmatrix} 1 & 0 \\ 0 & 1 \end{bmatrix} \tag{20}$$

In (20), we assume $\zeta \sim N(0,1)$.

Figure 3 shows the simulated target trajectory which is affected by model parametric uncertainties and its measurements with Gaussian noise. These measurements are used as the input of the standard KF and the proposed robust estimator. Figure 4 shows the estimation errors of the two methods. According to the comparison results in Figure 4, it can be seen that the proposed robust design effectively reduces the estimation error when the state is affected by model parametric uncertainties.

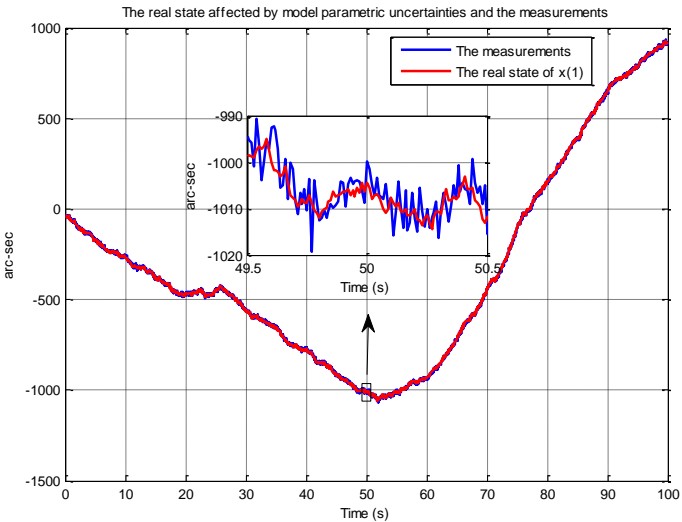

**Figure 3.** The real state affected by model parametric uncertainties and the measurements.

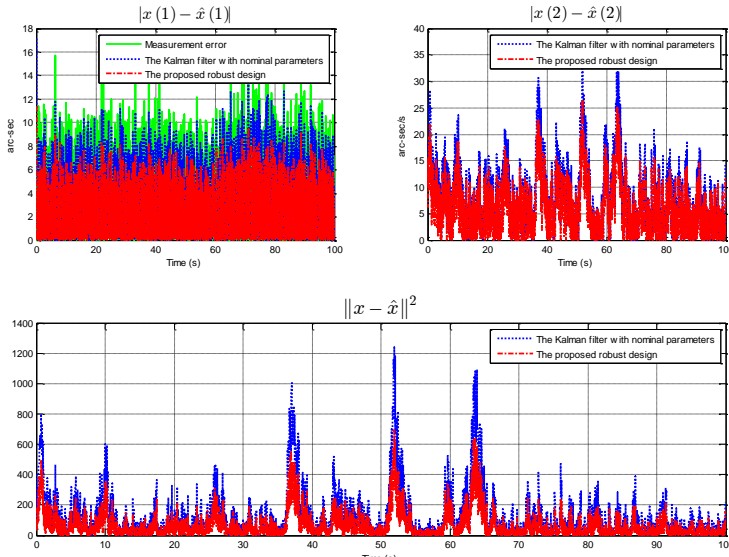

**Figure 4.** Comparison of estimation errors between SKF and the proposed robust design.

Through a comparative approach, $5 \times 10^2$ random simulations have been performed to demonstrate the effectiveness of the proposed estimator. Each simulation generates 1000 time-domain input/output data pairs for the state estimation of the plant, in which all initial states are set to 0, and the disturbances $w_k$ and $v_k$ are produced following normal

distributions. The ensemble-average estimated error variance of these $5 \times 10^2$ random simulations at each sampling time is calculated as follows:

$$E\left\|x_k - \hat{x}_{k|k}\right\|^2 \approx \frac{1}{500}\sum_{i=1}^{500}\left\|x_k - \hat{x}_{k|k}^{(i)}\right\|^2, \tag{21}$$

in which $i$ is the serial number of the random simulations.

Two other Kalman-based estimation methods are also simulated to compare with the proposed one in order to illustrate the effectiveness of the proposed method. One of them combines both the nominal parameters and the proposed augmented delay-free model (KFND). The other one combines with the actual parameters and the proposed augmented delay-free model (KFAD). Referencing the nominal parameters of [8,24,34], the system parameters selected in this example are as follows:

$$
\begin{aligned}
&A_k(\varepsilon_k) = \begin{bmatrix} 0.9802 & 0.0196 \\ 0.0000 & 0.9802 \end{bmatrix}, B_k(\varepsilon_k) = \begin{bmatrix} 1.0000 & 0.0000 \\ 0.0000 & 1.0000 \end{bmatrix}, \\
&C_k(\varepsilon_k) = \begin{bmatrix} 0.5000 + p \cdot \varepsilon_k & 0 \end{bmatrix}, R_k = 1.0000, \\
&Q_k = \begin{bmatrix} 1.9608 & 0.0195 \\ 0.0195 & 1.9605 \end{bmatrix}, \Pi_0 = \begin{bmatrix} 1.0000 & 0.0000 \\ 0.0000 & 1.0000 \end{bmatrix}
\end{aligned} \tag{22}
$$

In which the sampling period is $T = 0.01$ and the measurement delay frames of the system is $d = 3$. Besides, $p \in [0,1]$ and $\varepsilon_k \sim N(0,1)$.

The convergence properties of the example could be confirmed by the given theorems before the experiment. Definitions in (4) and (11), together with the determined distribution of $\varepsilon_k$, exhibit that $\bar{A}_k, \bar{B}_k, \bar{C}_k, R_k, Q_k, H_{k1}, H_{k2}$ are time-invariant. According to [37], the detectability of $(A, C)$ is equal to that if $\mathrm{Re}(\lambda_i) \geq 0$, then $rank[col(A - \lambda_i I, C)] = n$, where $\lambda_i$ is the eigenvalue of $A$. In the example of this article, $\lambda_1 = \lambda_2 = 0.9802$, and

$$rank\left(\begin{bmatrix} A - 0.9802I \\ C \end{bmatrix}\right) = rank\left(\begin{bmatrix} 0 & 0.0196 \\ 0 & 0 \\ 0.5 & 0 \end{bmatrix}\right) = 2 = n.$$

Therefore, $(A, C)$ is detectable. Similarly, the stabilization of $(M1, M2)$ has similar equivalent conditions. That is, if $\mathrm{Re}(\lambda_i) \geq 0$, then $rank\begin{bmatrix} M1 - \lambda_i I & M2 \end{bmatrix} = n$, where $\lambda_i$ is the eigenvalue of $M1$. It can be further verified by direct algebraic operations that $(M1, M2)$ is stabilizable. From the conditions required by Theorem 2, it can be proved that the estimator proposed in this paper is asymptotically stable when applied to the example in this section. Obviously, $\bar{A}_k, \bar{B}_k, \bar{C}_k, R_k, Q_k$ are all bounded for $k > 0$ and $\varepsilon_k \in \mathrm{E}$. Simple algebraic operations show that the eigenvalues of System (3) can be obtained as $\lambda_1 = \lambda_2 = \lambda_3 = 0, \lambda_4 = \lambda_5 = 0.9802$. They are all inside of the unit circle. With reference to [38], it can be drawn that System 3 is exponentially Lyapunov stable. On the basis of Theorem 2, the estimation error matrix of the proposed estimator of System (22) is confirmed to be bounded.

Next, the simulation is divided into two cases: Case 1. In these $5 \times 100$ simulations, the modelling errors $\varepsilon_k$ follow a normal distribution($\varepsilon_k \sim N(0,1)$). However, in the same simulation, $\varepsilon_k$ at each moment does not change. Case 2. In the same simulation, the modelling errors $\varepsilon_k$ at each moment follow a normal distribution($\varepsilon_k \sim N(0,1)$). For the two cases mentioned above, three sets of experiments are made. The differences among these three sets of experiments are the change of $p$. The purpose is to change the "size" of uncertainty. $p = 0.1$ is in the first group, $p = 0.5$, and $p = 1$ are in the second and third group, respectively.

Figures 5 and 6 show the simulation results of Case 1 and Case 2, respectively. According to the experimental results, the proposed method is robust to model parametric uncertainties. Especially when the uncertainty is "large", the contrast is more obvious. From the third group of experiments in Case 1, it can be concluded that the estimation error of the proposed method is about 50% lower than that of the KFND method. The

result of Case 2 shows that even if the model changes all the time, the proposed method is still robust. As the uncertainty decreases, the estimated performance of the three methods tends to be a similar level. This is because the method proposed in this article is an improvement on Kalman filter's RLS framework. When the uncertainty is 0, the proposed method degenerates into a standard Kalman filter.

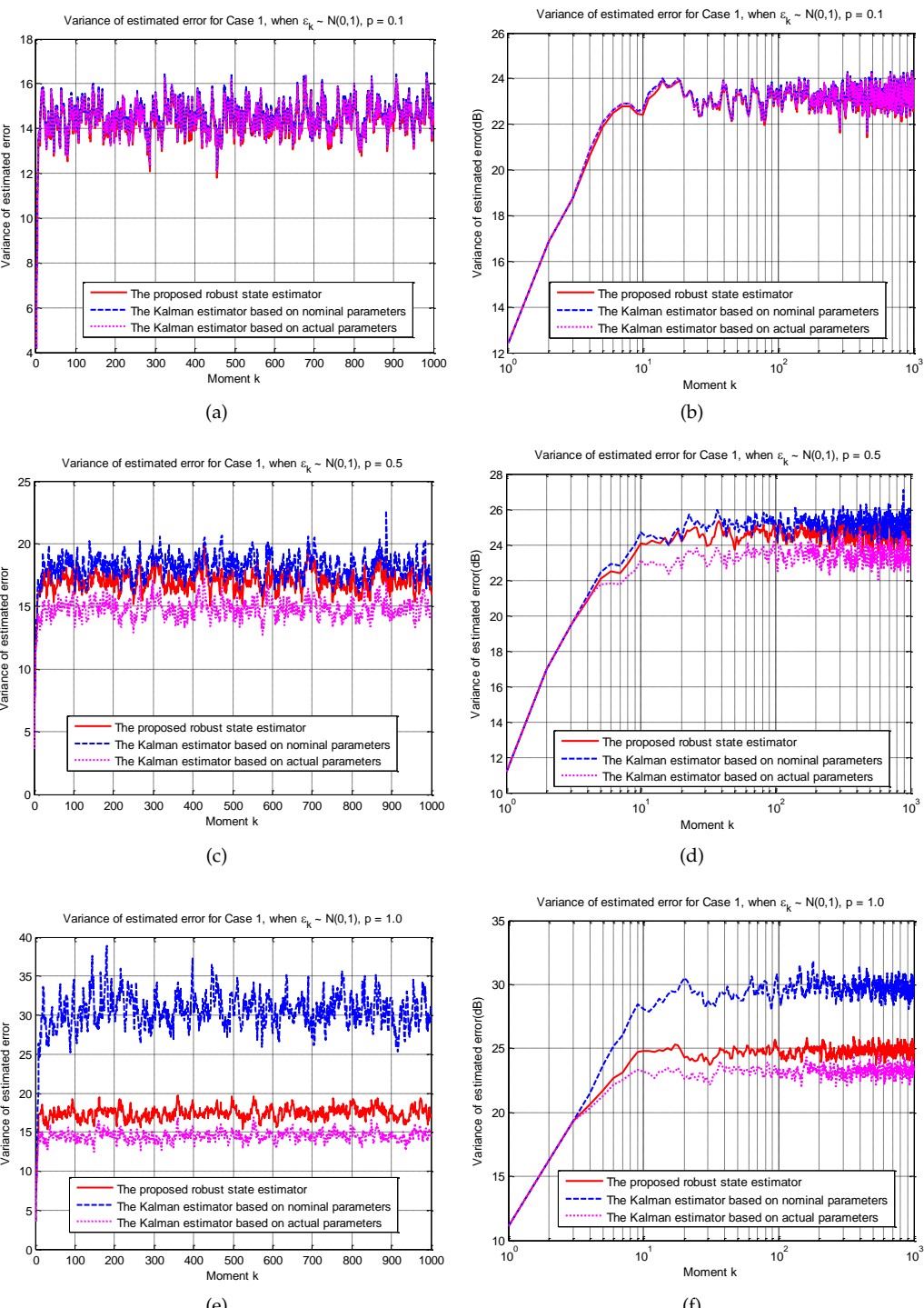

**Figure 5.** The experimental results of Case 1.

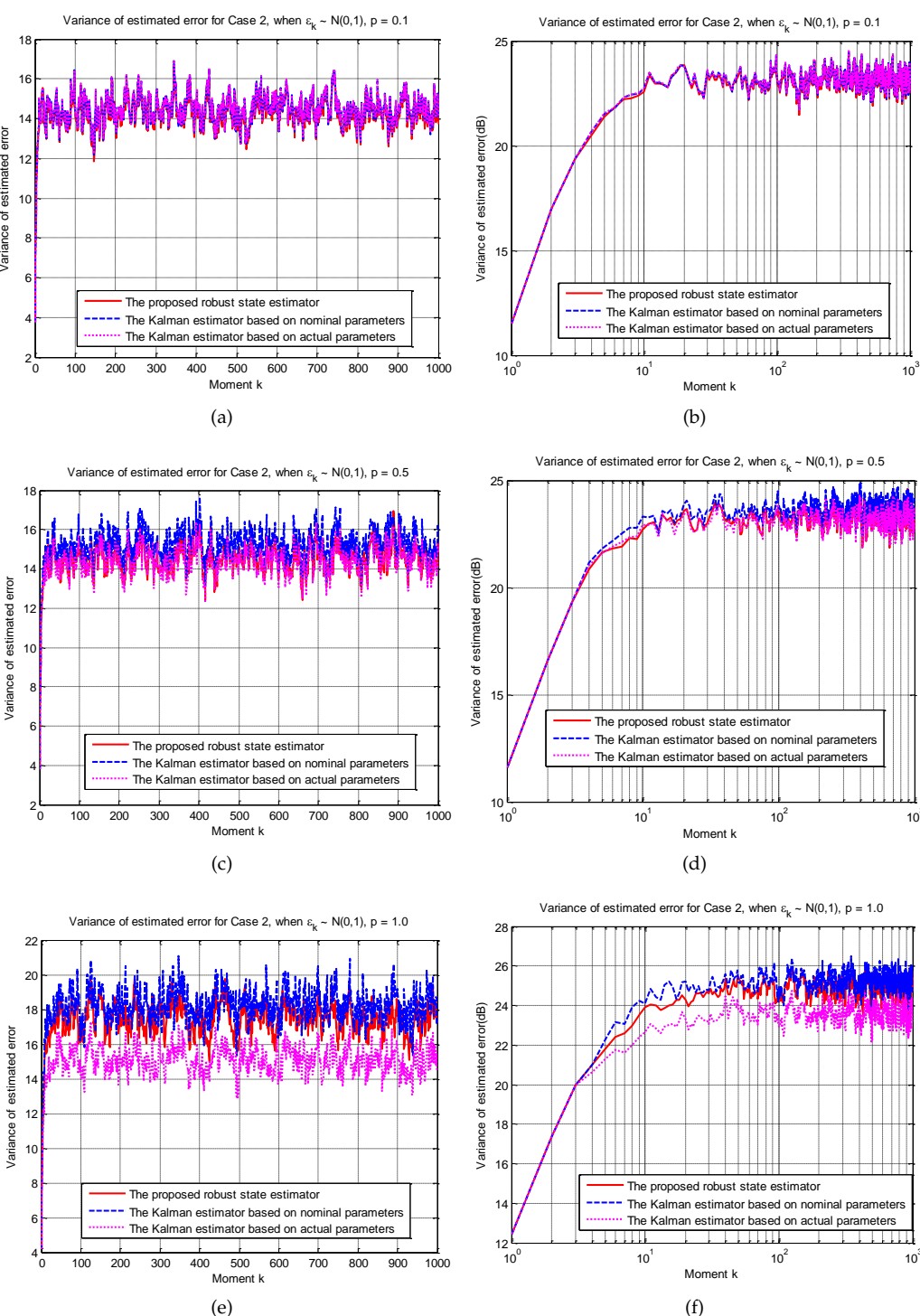

**Figure 6.** The experimental results of Case 2.

## 5. Conclusions

Aiming at the simultaneous existence of constant measurement delay and random parametric uncertainties in discrete-time linear systems, this paper proposes a new robust state estimator based on the combination of state augmentation and the improved Kalman filter RLS framework. A recursive filtering procedure similar to the Kalman filter is derived. The conditions for the asymptotic stability of the proposed estimator as well as the conditions for the boundedness of the estimation error matrix are explicitly given. The experimental results manifest that the robust estimator performs excellently, especially

when the system encounters a "large" uncertainty. At the same time, the robust estimator is still trustworthy even in the severe case that the system matrix changes at each moment. However, the computational cost of state augmentation makes the proposed estimator not conducive to the situation where the measurement delay is large. For nonlinear systems, the calculation of $H1$ and $H2$ will become more difficult. Both the asymptotic stability conditions of the estimator and the boundedness conditions of the error covariance need to be re-derived. Further research is needed in the future.

**Author Contributions:** Conceptualization, Z.L. and Y.M.; methodology, Z.L.; software, Z.L. and M.S.; validation, M.S.; formal analysis, Z.L.; investigation, M.S. and Q.D.; resources, Y.M.; data curation, Z.L. and Q.D.; writing—original draft preparation, Z.L.; writing—review and editing, Z.L. and M.S.; visualization, Q.D.; supervision, Y.M.; project administration, Y.M.; funding acquisition, Y.M. All authors have read and agreed to the published version of the manuscript.

**Funding:** This work was supported in part by the National Natural Science Foundation of China under Grant 61733012 and Grant 61905253. The authors have no relevant financial or non-financial interests to disclose.

**Institutional Review Board Statement:** Not applicable.

**Informed Consent Statement:** Not applicable.

**Data Availability Statement:** Not applicable.

**Conflicts of Interest:** The authors declare no conflict of interest. The funders had no role in the design of the study, collection, analyses, interpretation of data, writing of the manuscript, and in the decision to publish the results.

## Appendix A. Derivation of the Recursive Procedure

The first is the initial state estimation. There is no process noise at moment 0, the cost function of the estimator can be expressed as

$$J_0 = E\left\{ \|\bar{x}_0\|^2_{\bar{\Pi}_0^{-1}} + \|y_0 - \bar{C}_0(\varepsilon_0)\bar{x}_0\|^2_{R_0^{-1}} \right\} \tag{A1}$$

Find the partial derivative of $J_0$ for $\bar{x}_0$ and let $\partial J_0/\partial \bar{x}_0 = 0$. (A2) is obtained.

$$\hat{x}_{0|0} = \left( \bar{\Pi}_0^{-1} + E\left\{ \bar{C}_0^T(\varepsilon_0)R_0^{-1}\bar{C}_0(\varepsilon_0) \right\} \right)^{-1} E\left\{ \bar{C}_0(\varepsilon_0) \right\} R_0^{-1} y_0 \tag{A2}$$

where $\bar{\Pi}_0$ is the initial estimated variance of $\bar{x}_0$.

**Lemma A1.** *For any four matrices A, B, C, and D that have appropriate dimensions, assuming that the inverse of the correlation matrix exists, the following conclusions can be drawn [39]:*

$$\begin{bmatrix} A & B \\ C & D \end{bmatrix} = \begin{bmatrix} I & 0 \\ CA^{-1} & I \end{bmatrix} \begin{bmatrix} A & 0 \\ 0 & D - CA^{-1}B \end{bmatrix} \begin{bmatrix} I & A^{-1}B \\ 0 & I \end{bmatrix}$$
$$A(I + BA)^{-1} = (I + AB)^{-1}A \tag{A3}$$
$$(A + BCD)^{-1} = A^{-1} - A^{-1}B(DA^{-1}B + C^{-1})^{-1}DA^{-1}$$

According to Lemma A1 and (15), (A4) is obtained.

$$\left( \begin{bmatrix} P_{k|k}^{-1} & 0 \\ 0 & Q_k^{-1} \end{bmatrix} + G_k \right) = \left( \begin{bmatrix} P_{k|k}^{-1} & 0 \\ 0 & Q_k^{-1} \end{bmatrix} + \begin{bmatrix} G_{k11} & G_{k12} \\ G_{k12}^T & G_{k22} \end{bmatrix} \right)$$
$$= \begin{bmatrix} I & 0 \\ G_{k12}^T \hat{P}_{k|k} & I \end{bmatrix} \begin{bmatrix} \hat{P}_{k|k}^{-1} & 0 \\ 0 & U_k^{-1} \end{bmatrix} \begin{bmatrix} I & \hat{P}_{k|k} G_{k12} \\ 0 & I \end{bmatrix} \tag{A4}$$

Then, (A5) can be gained by substituting (A4) into (10),

$$
\begin{aligned}
H_{k2}R_{k+1}^{-1}y_{k+1} - H_{k3}\hat{x}_{k|k} &= \left( diag\left( P_{k|k}^{-1}, Q_k^{-1} \right) + G_k + col\left[ \bar{A}_k^T(0), \bar{B}_k^T(0) \right] \right. \\
&\quad \left. \times \bar{C}_{k+1}^T(0)R_{k+1}^{-1}\bar{C}_{k+1}(0)\left[ \bar{A}_k(0) \quad \bar{B}_k(0) \right] \right) col\left( \hat{x}_{k|k+1} - \hat{x}_{k|k}, \hat{w}_{k+1} \right) \\
&= \left( \begin{bmatrix} I & 0 \\ G_{k12}^T\hat{P}_{k|k} & I \end{bmatrix} \begin{bmatrix} \hat{P}_{k|k}^{-1} & 0 \\ 0 & U_k^{-1} \end{bmatrix} \begin{bmatrix} I & \hat{P}_{k|k}G_{k12} \\ 0 & I \end{bmatrix} + col\left[ \bar{A}_k^T(0), \bar{B}_k^T(0) \right] \right. \\
&\quad \left. \times \bar{C}_{k+1}^T(0)R_{k+1}^{-1}\bar{C}_{k+1}(0)\left[ \bar{A}_k(0) \quad \bar{B}_k(0) \right] \right) col\left( \hat{x}_{k|k+1} - \hat{x}_{k|k}, \hat{w}_{k+1} \right).
\end{aligned}
\tag{A5}
$$

Pre-multiply the matrix $\begin{bmatrix} I & 0 \\ -G_{k12}^T\hat{P}_{k|k} & I \end{bmatrix}$ by the left side of (A5) and $\begin{bmatrix} I & 0 \\ G_{k12}^T\hat{P}_{k|k} & I \end{bmatrix}^{-1}$ by the right side. The following derivations are obtained:

$$
\begin{aligned}
&\begin{bmatrix} \hat{P}_{k|k}^{-1} & 0 \\ 0 & U_k^{-1} \end{bmatrix} \begin{bmatrix} I & \hat{P}_{k|k}G_{k12} \\ 0 & I \end{bmatrix} \begin{pmatrix} \hat{\tilde{x}}_{k|k+1} - \hat{\tilde{x}}_{k|k} \\ \hat{w}_{k+1} \end{pmatrix} + \begin{bmatrix} I & 0 \\ -G_{k12}^T\hat{P}_{k|k} & I \end{bmatrix} \\
&\quad \times \begin{bmatrix} \bar{A}_k^T(0) \\ \bar{B}_k^T(0) \end{bmatrix} \bar{C}_{k+1}^T(0)R_{k+1}^{-1}\bar{C}_{k+1}(0)\left[ \bar{A}_k(0) \quad \bar{B}_k(0) \right] \begin{pmatrix} \hat{\tilde{x}}_{k|k+1} - \hat{\tilde{x}}_{k|k} \\ \hat{w}_{k+1} \end{pmatrix} \\
&= \begin{bmatrix} \hat{P}_{k|k}^{-1}\left( \hat{\tilde{x}}_{k|k+1} - \hat{\tilde{x}}_{k|k} \right) + G_{k12}\hat{w}_{k+1} \\ U_k^{-1}\hat{w}_{k+1} \end{bmatrix} + \begin{bmatrix} \bar{A}_k^T(0) \\ -G_{k12}^T\hat{P}_{k|k}\bar{A}_k^T(0) + \bar{B}_k^T(0) \end{bmatrix} \\
&\quad \times \bar{C}_{k+1}^T(0)R_{k+1}^{-1}\bar{C}_{k+1}(0)\left[ \bar{A}_k(0) \quad \bar{B}_k(0) - \bar{A}_k(0)\hat{P}_{k|k}G_{k12} \right] \\
&\quad \times \begin{pmatrix} \hat{\tilde{x}}_{k|k+1} + \hat{P}_{k|k}G_{k12}\hat{w}_{k+1} - \hat{\tilde{x}}_{k|k} \\ \hat{w}_{k+1} \end{pmatrix} \\
&= \begin{bmatrix} \hat{P}_{k|k}^{-1}\left( \hat{\tilde{x}}_{k|k+1} + \hat{P}_{k|k}G_{k12}\hat{w}_{k+1} - \hat{\tilde{x}}_{k|k} \right) \\ U_k^{-1}\hat{w}_{k+1} \end{bmatrix} \\
&\quad + \begin{bmatrix} \bar{A}_k^T(0) \\ \hat{B}_k^T(0) \end{bmatrix} \bar{C}_{k+1}^T(0)R_{k+1}^{-1}\bar{C}_{k+1}(0) \\
&\quad \times \left[ \bar{A}_k(0) \quad \bar{B}_k(0) \right] \begin{pmatrix} \hat{\tilde{x}}_{k|k+1} + \hat{P}_{k|k}G_{k12}\hat{w}_{k+1} - \hat{\tilde{x}}_{k|k} \\ \hat{w}_{k+1} \end{pmatrix} \\
&= \begin{bmatrix} \hat{P}_{k|k}^{-1} & 0 \\ 0 & U_k^{-1} \end{bmatrix} \begin{pmatrix} \tilde{\tilde{x}}_{k|k+1} - \hat{\tilde{x}}_{k|k} \\ \hat{w}_{k+1} \end{pmatrix} + \hat{H}_k^T\psi_k\hat{H}_k \begin{pmatrix} \tilde{\tilde{x}}_{k|k+1} - \hat{\tilde{x}}_{k|k} \\ \hat{w}_{k+1} \end{pmatrix} \\
&= \left( \begin{bmatrix} \hat{P}_{k|k}^{-1} & 0 \\ 0 & U_k^{-1} \end{bmatrix} + \hat{H}_k^T\psi_k\hat{H}_k \right) \begin{pmatrix} \tilde{\tilde{x}}_{k|k+1} - \hat{\tilde{x}}_{k|k} \\ \hat{w}_{k+1} \end{pmatrix} \\
&= \begin{bmatrix} I & 0 \\ -G_{k12}^T\hat{P}_{k|k} & I \end{bmatrix} \left( H_{k2}R_{k+1}^{-1}y_{k+1} - H_{k3}\hat{\tilde{x}}_{k|k} \right)
\end{aligned}
$$

That is,

$$
\begin{aligned}
&\left( \begin{bmatrix} \hat{P}_{k|k}^{-1} & 0 \\ 0 & U_k^{-1} \end{bmatrix} + \hat{H}_k^T\psi_k\hat{H}_k \right) \begin{pmatrix} \tilde{\tilde{x}}_{k|k+1} - \hat{\tilde{x}}_{k|k} \\ \hat{w}_{k+1} \end{pmatrix} \\
&= \begin{bmatrix} I & 0 \\ -G_{k12}^T\hat{P}_{k|k} & I \end{bmatrix} \left( H_{k2}R_{k+1}^{-1}y_{k+1} - H_{k3}\hat{\tilde{x}}_{k|k} \right)
\end{aligned}
\tag{A6}
$$

where $\tilde{\tilde{x}}_{k|k+1} = \hat{\tilde{x}}_{k|k+1} + \hat{P}_{k|k}G_{k12}\hat{w}_{k+1}$, $\hat{H}_k = \bar{C}_{k+1}(0)\left[\bar{A}_k(0) \quad \hat{B}_k(0)\right]$ and $\tilde{\tilde{x}}_{k+1|k+1} = \bar{A}_k(0)\tilde{\tilde{x}}_{k|k+1} + \hat{B}_k(0)\hat{w}_{k+1}$. The derivation in (A7) is obtained by taking out the first row of (A6).

$$
\begin{aligned}
&\hat{P}_{k|k}^{-1}\left(\tilde{\tilde{x}}_{k|k+1} - \hat{\tilde{x}}_{k|k}\right) + \bar{A}_k^T(0)\bar{C}_{k+1}^T(0)R_{k+1}^{-1} \\
&\times \bar{C}_{k+1}(0)\bar{A}_k(0)\left(\tilde{\tilde{x}}_{k|k+1} - \hat{\tilde{x}}_{k|k}\right) \\
&+ \bar{A}_k^T(0)\bar{C}_{k+1}^T(0)R_{k+1}^{-1}\bar{C}_{k+1}(0)\hat{B}_k(0)\hat{w}_{k+1} \\
&= \begin{bmatrix} I & 0 \end{bmatrix}\left(H_{k2}R_{k+1}^{-1}y_{k+1} - H_{k3}\hat{\tilde{x}}_{k|k}\right) \\
&\qquad\qquad\qquad \Downarrow \\
&\tilde{\tilde{x}}_{k|k+1} = \hat{\tilde{x}}_{k|k} + \hat{P}_{k|k}\begin{bmatrix} I & 0 \end{bmatrix}\left(H_{k2}R_{k+1}^{-1}y_{k+1} - H_{k3}\hat{\tilde{x}}_{k|k}\right) \\
&- \hat{P}_{k|k}\bar{A}_k^T(0)\bar{C}_{k+1}^T(0)R_{k+1}^{-1}\bar{C}_{k+1}(0)\left(\tilde{\tilde{x}}_{k+1|k+1} - \bar{A}_k(0)\hat{\tilde{x}}_{k|k}\right).
\end{aligned}
\tag{A7}
$$

In a similar fashion, the second row is taken out to obtain (A8).

$$
\begin{aligned}
&U_k^{-1}\hat{w}_{k+1} + \hat{B}_k^T(0)\bar{C}_{k+1}^T(0)R_{k+1}^{-1}\bar{C}_{k+1}(0)\bar{A}_k(0) \\
&\times \left(\tilde{\tilde{x}}_{k|k+1} - \hat{\tilde{x}}_{k|k}\right) + \hat{B}_k^T(0)\bar{C}_{k+1}^T(0)R_{k+1}^{-1}\bar{C}_{k+1}(0)\hat{B}_k^T(0)\hat{w}_{k+1} \\
&= \begin{bmatrix} -G_{k12}^T\hat{P}_{k|k} & I \end{bmatrix}\left(H_{k2}R_{k+1}^{-1}y_{k+1} - H_{k3}\hat{\tilde{x}}_{k|k}\right) \\
&\qquad\qquad\qquad \Downarrow \\
&\hat{w}_{k+1} = U_k\begin{bmatrix} -G_{k12}^T\hat{P}_{k|k} & I \end{bmatrix}\left(H_{k2}R_{k+1}^{-1}y_{k+1} - H_{k3}\hat{\tilde{x}}_{k|k}\right) \\
&- U_k\hat{B}_k^T(0)\bar{C}_{k+1}^T(0)R_{k+1}^{-1}\bar{C}_{k+1}(0)\left(\tilde{\tilde{x}}_{k+1|k+1} - \bar{A}_k(0)\hat{\tilde{x}}_{k|k}\right)
\end{aligned}
\tag{A8}
$$

From (A7) and (A8), the relationship between $\tilde{\tilde{x}}_{k+1|k+1}$ and $\tilde{\tilde{x}}_{k|k+1}$, as well as the relationship between $\tilde{\tilde{x}}_{k+1|k+1}$ and $\hat{w}_{k+1}$, are obtained. Then, substituting the results of (A7) and (A8) into $\tilde{\tilde{x}}_{k+1|k+1} = \bar{A}_k(0)\tilde{\tilde{x}}_{k|k+1} + \hat{B}_k(0)\hat{w}_{k+1}$, (A9) is obtained.

$$
\begin{aligned}
&\tilde{\tilde{x}}_{k+1|k+1} = \bar{A}_k(0)\tilde{\tilde{x}}_{k|k+1} + \hat{B}_k(0)\hat{w}_{k+1} \\
&= \bar{A}_k(0)\left\{\hat{\tilde{x}}_{k|k} + \hat{P}_{k|k}\begin{bmatrix} I_{n+md} & 0_{(n+md)\times n} \end{bmatrix}\left(H_{k2}R_{k+1}^{-1}y_{k+1} - H_{k3}\hat{\tilde{x}}_{k|k}\right)\right. \\
&\left. - \hat{P}_{k|k}\bar{A}_k^T(0)\bar{C}_{k+1}^T(0)R_{k+1}^{-1}\bar{C}_{k+1}(0)\left(\tilde{\tilde{x}}_{k+1|k+1} - \bar{A}_k(0)\hat{\tilde{x}}_{k|k}\right)\right\} \\
&+ \hat{B}_k(0)\left\{U_k\begin{bmatrix} -G_{k12}^T\hat{P}_{k|k} & I_n \end{bmatrix}\left(H_{k2}R_{k+1}^{-1}y_{k+1} - H_{k3}\hat{\tilde{x}}_{k|k}\right)\right. \\
&\left. - U_k\hat{B}_k^T(0)\bar{C}_{k+1}^T(0)R_{k+1}^{-1}\bar{C}_{k+1}(0)\left(\tilde{\tilde{x}}_{k+1|k+1} - \bar{A}_k(0)\hat{\tilde{x}}_{k|k}\right)\right\} \\
&= \left\{\bar{A}_k(0)\hat{P}_{k|k}\begin{bmatrix} I_{n+md} & 0_{(n+md)\times n} \end{bmatrix} + \hat{B}_k^T(0)U_k\begin{bmatrix} -G_{k12}^T\hat{P}_{k|k} & I_n \end{bmatrix}\right\}H_{k2}R_{k+1}^{-1}y_{k+1} \\
&- P_{k+1|k}\bar{C}_{k+1}^T(0)R_{k+1}^{-1}\bar{C}_{k+1}(0)\tilde{\tilde{x}}_{k+1|k+1} + \hat{\bar{A}}_k(0)\hat{\tilde{x}}_{k|k}
\end{aligned}
\tag{A9}
$$

For further simplifying (A9),

$$
\begin{aligned}
\tilde{\tilde{x}}_{k+1|k+1} &= \left(I + P_{k+1|k}\bar{C}_{k+1}^T(0)R_{k+1}^{-1}\bar{C}_{k+1}(0)\right)^{-1} \\
&\times \left\{\left(\bar{A}_k(0)\hat{P}_{k|k}\begin{bmatrix} I & 0 \end{bmatrix} + \hat{B}_k^T(0)U_k\begin{bmatrix} -G_{k12}^T\hat{P}_{k|k} & I \end{bmatrix}\right)\right. \\
&\left. \times H_{k2}R_{k+1}^{-1}y_{k+1} + \hat{\bar{A}}_k(0)\hat{\tilde{x}}_{k|k}\right\}
\end{aligned}
\tag{A10}
$$

is obtained where

$$
\begin{aligned}
&\left(I + P_{k+1|k}\bar{C}_{k+1}^T(0)R_{k+1}^{-1}\bar{C}_{k+1}(0)\right)^{-1} \\
&= I^{-1} - I^{-1}P_{k+1|k}\bar{C}_{k+1}^T(0) \\
&\quad \times \left(\bar{C}_{k+1}(0)I^{-1}P_{k+1|k}\bar{C}_{k+1}^T(0) + R_{k+1}\right)^{-1}\bar{C}_{k+1}(0)I^{-1} \\
&= I - P_{k+1|k}\bar{C}_{k+1}^T(0)\left(R_{k+1} + \bar{C}_{k+1}(0)P_{k+1|k}\bar{C}_{k+1}^T(0)\right)^{-1}\bar{C}_{k+1}(0)
\end{aligned}
\tag{A11}
$$

according to Lemma A1. Then, (A11) becomes (A12) according to the definitions in (16).

$$
\begin{aligned}
&\left(I + P_{k+1|k}\bar{C}_{k+1}^T(0)R_{k+1}^{-1}\bar{C}_{k+1}(0)\right)^{-1}P_{k+1|k} \\
&= \left(I - P_{k+1|k}\bar{C}_{k+1}^T(0)R_{e,k+1}^{-1}\bar{C}_{k+1}(0)\right)P_{k+1|k} \\
&= P_{k+1|k+1}
\end{aligned}
\tag{A12}
$$

Combining Lemma A1 and (16) to inverse matrix $P_{k+1|k+1}$,

$$
\begin{aligned}
P_{k+1|k+1}^{-1} &= \left[\left(I - P_{k+1|k}\bar{C}_{k+1}^T(0)R_{e,k+1}^{-1}\bar{C}_{k+1}(0)\right)P_{k+1|k}\right]^{-1} \\
&= P_{k+1|k}^{-1}\left(I - P_{k+1|k}\bar{C}_{k+1}^T(0)R_{e,k+1}^{-1}\bar{C}_{k+1}(0)\right)^{-1} \\
&= P_{k+1|k}^{-1}\left(I + P_{k+1|k}\bar{C}_{k+1}^T(0)R_{k+1}^{-1}\bar{C}_{k+1}(0)\right) \\
&= P_{k+1|k}^{-1} + \bar{C}_{k+1}^T(0)R_{k+1}^{-1}\bar{C}_{k+1}(0)
\end{aligned}
\tag{A13}
$$

is obtained. Further, (A14) is obtained by combining (A12) and (A13).

$$
\begin{aligned}
&\left(I - P_{k+1|k+1}\bar{C}_{k+1}^T(0)R_{k+1}^{-1}\bar{C}_{k+1}(0)\right) \\
&= P_{k+1|k+1}\left(P_{k+1|k+1}^{-1} - \bar{C}_{k+1}^T(0)R_{k+1}^{-1}\bar{C}_{k+1}(0)\right) \\
&= P_{k+1|k+1}\left(P_{k+1|k}^{-1} + \bar{C}_{k+1}^T(0)R_{k+1}^{-1}\bar{C}_{k+1}(0) - \bar{C}_{k+1}^T(0)R_{k+1}^{-1}\bar{C}_{k+1}(0)\right) \\
&= P_{k+1|k+1}P_{k+1|k}^{-1} \\
&= \left(I + P_{k+1|k}\bar{C}_{k+1}^T(0)R_{k+1}^{-1}\bar{C}_{k+1}(0)\right)^{-1}
\end{aligned}
\tag{A14}
$$

Substituting (A12)–(A14) into (A10), (A10) is further simplified to

$$
\begin{aligned}
\tilde{x}_{k+1|k+1} &= \hat{A}_k(0)\hat{x}_{k|k} + P_{k+1|k+1} \\
&\quad \times \left(P_{k+1|k}^{-1}\left(\bar{A}_k(0)\hat{P}_{k|k}\begin{bmatrix} I_{n+md} & 0_{(n+md)\times n}\end{bmatrix}\right.\right. \\
&\quad + \hat{B}_k(0)U_k\hat{B}_k^T(0)\begin{bmatrix} -G_{k12}^T\hat{P}_{k|k} & I_n\end{bmatrix}\right)H_{k2}R_{k+1}^{-1}y_{k+1} \\
&\quad \left.- \bar{C}_{k+1}^T(0)R_{k+1}^{-1}\bar{C}_{k+1}(0)\hat{A}_k(0)\hat{x}_{k|k}\right).
\end{aligned}
\tag{A15}
$$

Note that the definition of $\tilde{x}_{k+1|k+1} = \bar{A}_k(0)\tilde{x}_{k|k+1} + \hat{B}_k(0)\hat{w}_{k+1}$ is similar to [8,24,34], which means that $\tilde{x}_{k+1|k+1}$ can be designated as $\hat{x}_{k+1|k+1}$. The derivation of the recursive procedure is complete.

## Appendix B

Proof of Theorem 1.

**Lemma A2.** *If $(A, C)$ is detectable and $rank(A) = n$, then $(\bar{A}_k, F_k)$ is detectable.*

**Proof.** Suppose $\bar{A}_k v_i = \lambda_i v_i$, $\lambda_i$ is the eigenvalue of $\bar{A}_k$ and $v_i$ is the eigenvector of $\bar{A}_k$. According to [37], an equivalent condition for $(\bar{A}_k, F_k)$ to be detectable is that if $Re(\lambda_i) \geq 0$, then $rank[col(\bar{A}_k - \lambda_i I, F_k)] = n + dm$. According to the definitions of $\bar{A}_k$ and $F_k$ in Theorem 2,

$$
\begin{bmatrix} \bar{A}_k - \lambda_i I \\ F_k \end{bmatrix} = \begin{bmatrix} \begin{bmatrix} A - \lambda_i I_n & 0_{n \times dm} \\ \begin{bmatrix} C - CA \\ 0_{(d-1)m \times n} \end{bmatrix} & \begin{bmatrix} 0_{m \times (d-1)m} & 0_{m \times m} \\ I_{(d-1)m} & 0_{(d-1)m \times m} \end{bmatrix} - \lambda_i I_{dm} \end{bmatrix} \\ R_k^{-1/2} \cdot \begin{bmatrix} C_k, \overbrace{I_m, \cdots, I_m}^{d} \end{bmatrix} \\ G_{k11}^{1/2} \end{bmatrix}
$$

is obtained. Because the row rank is equal to column rank for any matrix, $rank[col(\bar{A}_k - \lambda_i I, F_k)] \leq n + dm$. Since $R_k \neq 0$, $R_k^{-1/2} \neq 0$. According to the expanded expression of $col(\bar{A}_k - \lambda_i I, F_k)$, (A16) is obtained.

$$
TR \triangleq rank \left( \begin{bmatrix} \begin{bmatrix} A - \lambda_i I_{n \times n} & 0_{n \times dm} \\ \begin{bmatrix} C - CA \\ 0_{(d-1)m \times n} \end{bmatrix} & \begin{bmatrix} 0_{m \times (d-1)m} & 0_{m \times m} \\ I_{(d-1)m} & 0_{(d-1)m \times m} \end{bmatrix} - \lambda_i I_{dm} \end{bmatrix} \\ R_k^{-1/2} \cdot \begin{bmatrix} C_k, \overbrace{I_m, \cdots, I_m}^{d} \end{bmatrix} \end{bmatrix} \right) \tag{A16}
$$

$$
\leq rank[col(\bar{A}_k - \lambda_i I, F_k)] \leq n + dm
$$

$\square$

If $\lambda_i = 0$, then $TR = rank[col(A, C - CA)] + (d - 1)m + m$. If $\lambda_i \neq 0$, then $TR = rank(A - \lambda_i I) + dm + m$.

Because $(A, C)$ is detectable and $rank(A) = n$, then $rank[col(A - \lambda_i I, C)] = n$ for any $\lambda_i$ satisfying $Re(\lambda_i) \geq 0$. Since $rank(C) \leq m$, then $rank(A - \lambda_i I) \geq n - m$. When $\lambda_i \neq 0$, (A17) is established.

$$
TR = rank(A - \lambda_i I) + dm + m \geq n + dm. \tag{A17}
$$

The conclusion of $rank[col(\bar{A}_k - \lambda_i I, F_k)] = n + dm$ can be drawn by combining (A16) and (A17) when $\lambda_i \neq 0$. Because $rank(A) = n$, $rank[col(A, C - CA)] = n + dm$. When $\lambda_i = 0$, (A18) is obtained.

$$
TR = rank[col(A, C - CA)] + (d - 1)m + m = n + dm \tag{A18}
$$

Combining (A18) and (A16), it can be concluded that when $\lambda_i = 0$, $rank[col(\bar{A}_k - \lambda_i I, F_k)] = n + dm$, the proof of Lemma A2 is complete.

According to Lemma A1 and the definition in (15), (A19)–(A21) can be obtained.

$$
\begin{aligned}
P_{k|k} &= P_{k|k-1} - P_{k|k-1}\bar{C}_k^T\left(R_k + \bar{C}_k P_{k|k-1}\bar{C}_k^T\right)\bar{C}_k P_{k|k-1}, \\
\hat{P}_{k|k} &= \left(P_{k|k}^{-1} + G_{k11}\right)^{-1} = \left(P_{k|k}^{-1} + G_{k11}^{1/2} I G_{k11}^{1/2}\right)^{-1} \\
&= P_{k|k} - P_{k|k} G_{k11}^{1/2}\left(G_{k11}^{1/2} P_{k|k} G_{k11}^{1/2} + I\right)^{-1} G_{k11}^{1/2} P_{k|k}, \\
U_k &= \left(Q_k^{-1} + G_{k22} - G_{k12}^T\left(P_{k|k}^{-1} + G_{k11}\right)^{-1} G_{k12}\right)^{-1} \\
&= \left(Q_k^{-1} + G_{k22} - G_{k12}^T G_{k11}^{-1} G_{k12} + G_{k12}^T G_{k11}^{-1/2}\right. \\
&\qquad \left.\times \left(I + G_{k11}^{1/2} P_{k|k} G_{k11}^{1/2}\right)^{-1} G_{k11}^{-1/2} G_{k12}\right)^{-1},
\end{aligned}
\tag{A19}
$$

$$
\begin{aligned}
\bar{B}_k U_k G_{k12}^T \hat{P}_{k|k}\bar{A}_k^T &= \bar{B}_k U_k\left(I + G_{k12}^T G_{k11}^{-1/2}\left(I + G_{k11}^{1/2} P_{k|k} G_{k11}^{1/2}\right)^{-1} G_{k11}^{-1/2} G_{k12} U_k\right)^{-1} \\
&\times G_{k12}^T G_{k11}^{-1/2}\left(I + G_{k11}^{1/2} P_{k|k} G_{k11}^{1/2}\right)^{-1} G_{k11}^{1/2} P_{k|k}\bar{A}_k^T = \bar{B}_k U_k G_{k12}^T G_{k11}^{-1/2}\left(\left(I + G_{k11}^{1/2} P_{k|k} G_{k11}^{1/2}\right)\right. \\
&\times \left.\left(I + \left(I + G_{k11}^{1/2} P_{k|k} G_{k11}^{1/2}\right)^{-1} G_{k11}^{-1/2} G_{k12} U_k G_{k12}^T G_{k11}^{-1/2}\right)\right)^{-1} \times G_{k11}^{1/2} P_{k|k}\bar{A}_k^T \\
&= \bar{B}_k U_k G_{k12}^T G_{k11}^{-1/2} \times \left(I + G_{k11}^{-1/2} G_{k12} U_k G_{k12}^T G_{k11}^{-1/2} + G_{k11}^{1/2} P_{k|k} G_{k11}^{1/2}\right)^{-1} \times G_{k11}^{1/2} P_{k|k}\bar{A}_k^T,
\end{aligned}
\tag{A20}
$$

$$
\begin{aligned}
&\bar{A}_k P_{k|k} G_{k11}^{1/2}\left(I + G_{k11}^{1/2} P_{k|k} G_{k11}^{1/2} + G_{k11}^{-1/2} G_{k12} U_k G_{k12}^T G_{k11}^{-1/2}\right)^{-1} \\
&\quad \times G_{k11}^{1/2} P_{k|k}\bar{A}_k^T \\
&= \bar{A}_k P_{k|k} G_{k11}^{1/2}\left(I + G_{k11}^{1/2} P_{k|k} G_{k11}^{1/2}\right)^{-1} G_{k11}^{1/2} P_{k|k}\bar{A}_k^T \\
&\quad - \bar{A}_k P_{k|k} G_{k11}^{1/2}\left(I + G_{k11}^{1/2} P_{k|k} G_{k11}^{1/2}\right)^{-1} \\
&\quad \times G_{k11}^{-1/2} G_{k12} U_k G_{k12}^T G_{k11}^{-1/2}\left(I + G_{k11}^{1/2} P_{k|k} G_{k11}^{1/2}\right)^{-1} G_{k11}^{1/2} P_{k|k}\bar{A}_k^T.
\end{aligned}
\tag{A21}
$$

According to the definitions of (15) and (16),

$$
\begin{aligned}
P_{k+1|k} &= \bar{A}_k\hat{P}_{k|k} A_k^T + \hat{B}_k U_k\hat{B}_k^T \\
&= \bar{A}_k\left(P_{k|k} - P_{k|k} G_{k11}^{1/2}\left(I + G_{k11}^{1/2} P_{k|k} G_{k11}^{1/2}\right)^{-1} G_{k11}^{1/2} P_{k|k}\right)\bar{A}_k^T \\
&\quad + \left(\bar{B}_k - \bar{A}_k\hat{P}_{k|k} G_{k12}\right) U_k\left(\bar{B}_k - \bar{A}_k\hat{P}_{k|k} G_{k12}\right)^T \\
&= \bar{A}_k P_{k|k}\bar{A}_k^T + \bar{B}_k U_k\bar{B}_k^T - \bar{B}_k U_k G_{k12}^T\hat{P}_{k|k}\bar{A}_k^T \\
&\quad - \bar{A}_k\hat{P}_{k|k} G_{k12} U_k\bar{B}_k^T - \bar{A}_k P_{k|k} G_{k11}^{1/2}\left(I + G_{k11}^{1/2} P_{k|k} G_{k11}^{1/2}\right)^{-1} G_{k11}^{1/2} P_{k|k}\bar{A}_k^T \\
&\quad + \bar{A}_k P_{k|k} G_{k11}^{1/2}\left(I + G_{k11}^{1/2} P_{k|k} G_{k11}^{1/2}\right)^{-1} G_{k11}^{-1/2} G_{k11} U_k G_{k12}^T G_{k11}^{-1/2}\left(I + G_{k11}^{1/2} P_{k|k} G_{k11}^{1/2}\right)^{-1} \\
&\quad \times G_{k11}^{1/2} P_{k|k}\bar{A}_k^T
\end{aligned}
\tag{A22}
$$

is obtained. Substituting (A20) and (A21) into $P_{k+1|k}$ and combining (A19), (A23) is obtained.

$$
\begin{aligned}
P_{k+1|k} &= \bar{A}_k P_{k|k-1} \bar{A}_k^T - \bar{A}_k P_{k|k} G_{k11}^{1/2} \\
&\times \left( I + G_{k11}^{-1/2} G_{k12} U_k G_{k12}^T G_{k11}^{-1/2} + G_{k11}^{1/2} P_{k|k} G_{k11}^{1/2} \right)^{-1} G_{k11}^{1/2} P_{k|k} \bar{A}_k^T \\
&- \bar{B}_k U_k G_{k12}^T G_{k11}^{-1/2} \left( I + G_{k11}^{-1/2} G_{k12} U_k G_{k12}^T G_{k11}^{-1/2} + G_{k11}^{1/2} P_{k|k} G_{k11}^{1/2} \right)^{-1} \\
&\times G_{k11}^{1/2} P_{k|k} \bar{A}_k^T - \bar{A}_k P_{k|k} G_{k11}^{1/2} \\
&\times \left( I + G_{k11}^{-1/2} G_{k12} U_k G_{k12}^T G_{k11}^{-1/2} + G_{k11}^{1/2} P_{k|k} G_{k11}^{1/2} \right)^{-1} \\
&\times G_{k11}^{-1/2} G_{k12} U_k \bar{B}_k^T - \bar{A}_k P_{k|k-1} \bar{C}_k^T R_k^{-1/2} (I + R_k^{-1/2} \bar{B}_k P_{k|k-1} \bar{C}_k R_k^{-1/2})^{-1} \\
&\times R_k^{-1/2} \bar{C}_k P_{k|k-1} \bar{A}_k^T - \bar{B}_k U_k G_{k12}^T G_{k11}^{-1/2} \\
&\times \left( I + G_{k11}^{-1/2} G_{k12} U_k G_{k12}^T G_{k11}^{-1/2} + G_{k11}^{1/2} P_{k|k} G_{k11}^{1/2} \right)^{-1} \\
&\times G_{k11}^{-1/2} G_{k12} U_k \bar{B}_k^T + \bar{B}_k U_k \bar{B}_k^T \\
&= \bar{A}_k P_{k|k-1} \bar{A}_k^T + \bar{B}_k U_k \bar{B}_k^T \\
&- \begin{bmatrix} \bar{A}_k P_{k|k-1} \bar{C}_k^T R_k^{-1/2} & \bar{A}_k P_{k|k} G_{k11}^{1/2} + \bar{B}_k U_k G_{k12}^T G_{k11}^{-1/2} \end{bmatrix} \\
&\times \begin{bmatrix} \left( I + R_k^{-1/2} \bar{C}_k P_{k|k-1} \bar{C}_k^T R_k^{-1/2} \right)^{-1} & 0 \\ 0 & \left( \begin{matrix} I + G_{k11}^{-1/2} G_{k12} U_k G_{k12}^T G_{k11}^{-1/2} \\ + G_{k11}^{1/2} P_{k|k} G_{k11}^{1/2} \end{matrix} \right)^{-1} \end{bmatrix} \\
&\times \begin{bmatrix} R_k^{-1/2} \bar{C}_k P_{k|k-1} \bar{A}_k^T \\ G_{k11}^{1/2} P_{k|k} \bar{A}_k^T + G_{k11}^{-1/2} G_{k12} U_k \bar{B}_k^T \end{bmatrix}
\end{aligned}
\tag{A23}
$$

Because of (A24), (A25) is obtained.

$$
\begin{aligned}
&\begin{bmatrix} \bar{A}_k P_{k|k-1} \bar{C}_k^T R_k^{-1/2} & \bar{A}_k P_{k|k} G_{k11}^{1/2} + \bar{B}_k U_k G_{k12}^T G_{k11}^{-1/2} \end{bmatrix} \\
&= \left( \bar{A}_k P_{k|k-1} F_k^T + D_k J_k \right) \\
&\times \begin{bmatrix} I & \left( I + R_k^{-1/2} \bar{C}_k P_{k|k-1} \bar{C}_k^T R_k^{-1/2} \right)^{-1} R_k^{-1/2} \bar{C}_k P_{k|k-1} G_{k11}^{1/2} \\ 0 & I \end{bmatrix}^{-1\prime}
\end{aligned}
\tag{A24}
$$

$$
\begin{aligned}
P_{k+1|k} &= \bar{A}_k P_{k|k-1} \bar{A}_k^T + \bar{B}_k U_k \bar{B}_k^T - \left( \bar{A}_k P_{k|k-1} F_k^T + D_k J_k \right) \\
&\times \begin{pmatrix} I + R_k^{-1/2} \bar{C}_k P_{k|k-1} \bar{C}_k^T R_k^{-1/2} & \left( I + R_k^{-1/2} \bar{C}_k P_{k|k-1} \bar{C}_k^T R_k^{-1/2} \right)^{-1} R_k^{-1/2} \bar{C}_k P_{k|k-1} G_{k11}^{1/2} \\ G_{k11}^{1/2} P_{k|k-1} C_k^T R_k^{-1/2} & I + G_{k11}^{-1/2} G_{k12} U_k G_{k12}^T G_{k11}^{-1/2} + G_{k11}^{1/2} P_{k|k} G_{k11}^{1/2} \end{pmatrix}^{-1} \\
&\times \left( \bar{A}_k P_{k|k-1} F_k^T + D_k J_k \right)^T \\
&= \bar{A}_k P_{k|k-1} \bar{A}_k^T + \bar{B}_k U_k \bar{B}_k^T - \left( \bar{A}_k P_{k|k-1} F_k^T + D_k J_k \right) \left( W_k + F_k P_{k|k-1} F_k^T \right)^{-1} \left( \bar{A}_k P_{k|k-1} F_k^T + D_k J_k \right)^T.
\end{aligned}
\tag{A25}
$$

(17) can be rewritten as

$$
\hat{x}_{k+1|k+1} = A_{fk} \hat{x}_{k|k} + P_{k+1|k+1} P_{k+1|k}^{-1} B_{fk} y_{k+1},
\tag{A26}
$$

in which,

$$A_{fk} = \left[I - P_{k+1|k+1}C_{k+1}^T(0)R_{k+1}^{-1}C_{k+1}(0)\right]\hat{A}_k(0)$$
$$B_{fk} = \left(\bar{A}_k(0)\hat{P}_{k|k}\left[I_{n+md} \quad 0_{(n+md)\times n}\right]\right.$$
$$\left.+\hat{B}_k(0)U_k\hat{B}_k^T(0)\left[-G_{k12}^T\hat{P}_{k|k} \quad I_n\right]\right)H_{k2}R_{k+1}^{-1}. \tag{A27}$$

From the relationship between $P_{k|k-1}$ and $P_{k|k}$ in (16), the convergence of $P_{k|k-1}$ is equivalent to $P_{k|k}$. Note that the last term of (A25) is a standard discrete Riccati algebraic equation. It follows the Theorem E.6.2 in [35]. That is, if $(\bar{A}_k, F_k)$ is detectable and $\left(\bar{A}_k - \bar{B}_k U_k G_{k12}^T(I + G_{k11}^1 G_{k12} U_k G_{k12}^T)^{-1}, D_k\left(I + U_k^{\frac{1}{2}}G_{k12}^T G_{k11}^1 G_{k12}U_k^{\frac{1}{2}}\right)^{-\frac{1}{2}}\right)$ is stabilizable, then $P_{k+1|k}$ has a unique positive-semi-definite solution. Combined with (A26), it can be seen that the above conditions are the asymptotic stability conditions of the proposed estimator too. Considering the conclusions of Lemma A2 and the relevant definitions in (4) and (18), the simplified conditions for a asymptotically stable estimator is obtained. Furthermore, the conditions are that $(A, C)$ is detectable, $rank(A) = n$ and $(M1, M2)$ is stabilizable.

Theorem 1 is proved.

**Appendix C**

Proof of Theorem 2.

First, define a matrix, as shown in (A28).

$$A_{pk} = \bar{A}_k - \left(\bar{A}_k P_{k|k-1}F_k^T + D_k J_k\right)\left(W_k + F_k P_{k|k-1}F_k^T\right)^{-1}F_k \tag{A28}$$

To simplify $A_{pk}$, it is necessary to know that the following two equations are true.

$$\bar{A}_k P_{k|k}G_{k11}^{1/2}\left(I + G_{k11}^{-1/2}G_{k12}U_k G_{k12}^T G_{k11}^{-1/2} + G_{k11}^{1/2}P_{k|k}G_{k11}^{1/2}\right)^{-1}G_{k11}^{1/2}$$
$$= \bar{A}_k\left(P_{k|k}^{-1} + G_{k11}\right)^{-1}G_{k11} - \bar{A}_k\left(P_{k|k}^{-1} + G_{k11}\right)^{-1}G_{k12}U_k G_{k12}^T\left(I + P_{k|k}G_{k11}\right)^{-1}$$
$$\bar{B}_k U_k G_{k12}^T G_{k11}^{-1/2}\left(I + G_{k11}^{-1/2}G_{k12}U_k G_{k12}^T G_{k11}^{-1/2}\right.$$
$$\left.+G_{k11}^{1/2}P_{k|k}G_{k11}^{1/2}\right)^{-1}G_{k11}^{1/2} = B_k U_k G_{k12}^T\left(I + P_{k|k}G_{k11}\right)^{-1} \tag{A29}$$

According to (18) and (A28), (A30) is obtained.

$$A_{pk} = \bar{A}_k - \left(\bar{A}_k P_{k|k-1}\left[\begin{array}{cc}\bar{C}_k^T R_k^{-1} & G_{k11}^{1/2}\end{array}\right] + \left[\begin{array}{cc}0 & \bar{B}_k Q_k G_{k12}^T G_{k11}^{-1/2}\end{array}\right]\right)$$
$$\times\left(\left[\begin{array}{cc}I & 0 \\ 0 & I + G_{k11}^{-1/2}G_{k12}U_k G_{k12}^T G_{k11}^{-1/2}\end{array}\right] + \left[\begin{array}{c}R_k^{-1/2}\bar{C}_k \\ G_{k11}^{1/2}\end{array}\right]P_{k|k-1}\left[\begin{array}{cc}\bar{C}_k^T R_k^{-1/2} & G_{k11}^{1/2}\end{array}\right]\right)^{-1}$$
$$= \bar{A}_k - \left(\bar{A}_k P_{k|k-1}\bar{C}_k^T R_k^{-1/2}\right)\times\left(I + R_k^{-1/2}\bar{C}_k P_{k|k-1}\bar{C}_k^T R_k^{-1/2}\right)^{-1}R_k^{-1/2}\bar{C}_k$$
$$-\left(\bar{A}_k P_{k|k-1}G_{k11}^{1/2} + \bar{B}_k Q_k G_{k12}^T G_{k11}^{-1/2}\right)$$
$$\times\left(I + G_{k11}^{-1/2}G_{k12}U_k G_{k12}^T G_{k11}^{-1/2} + G_{k11}^{-1/2}P_{k|k}G_{k11}^{1/2}\right)^{-1}G_{k11}^{1/2}\times\left(I + P_{k|k-1}\bar{C}_k^T R_k^{-1}\bar{C}_k\right)^{-1}$$
$$= \bar{A}_k\left(I + P_{k|k-1}\bar{C}_k^T R_k^{-1}\bar{C}_k\right)^{-1} - \left(\bar{A}_k P_{k|k-1}G_{k11}^{1/2} + \bar{B}_k Q_k G_{k12}^T G_{k11}^{-1/2}\right)$$
$$\times\left(I + G_{k11}^{-1/2}G_{k12}U_k G_{k12}^T G_{k11}^{-1/2} + G_{k11}^{-1/2}P_{k|k}G_{k11}^{1/2}\right)^{-1}\times G_{k11}^{1/2}\left(I + P_{k|k-1}\bar{C}_k^T R_k^{-1}\bar{C}_k\right)^{-1} \tag{A30}$$

Note that

$$
\begin{aligned}
& \bar{A}_k - \left( \bar{A}_k P_{k|k-1} G_{k11}^{1/2} + \bar{B}_k Q_k G_{k12}^T G_{k11}^{-1/2} \right) \\
& \times \left( I + G_{k11}^{-1/2} G_{k12} U_k G_{k12}^T G_{k11}^{-1/2} + G_{k11}^{-1/2} P_{k|k} G_{k11}^{1/2} \right)^{-1} G_{k11}^{1/2} \\
&= \bar{A}_k - \bar{A}_k P_{k|k-1} G_{k11}^{1/2} \left( I + G_{k11}^{-1/2} G_{k12} U_k G_{k12}^T G_{k11}^{-1/2} \right. \\
&\left. + G_{k11}^{-1/2} P_{k|k} G_{k11}^{1/2} \right)^{-1} G_{k11}^{1/2} - \bar{B}_k Q_k G_{k12}^T G_{k11}^{-1/2} \\
& \times \left( I + G_{k11}^{-1/2} G_{k12} U_k G_{k12}^T G_{k11}^{-1/2} + G_{k11}^{-1/2} P_{k|k} G_{k11}^{1/2} \right)^{-1} G_{k11}^{1/2} \\
&= \bar{A}_k - \bar{A}_k P_{k|k-1} G_{k11} + \bar{A}_k \hat{P}_{k|k} G_{k12} U_k G_{k12}^T \left( I - \hat{P}_{k|k} G_{k11} \right) \\
& - \bar{B}_k U_k G_{k12}^T \left( I - \hat{P}_{k|k} G_{k11} \right) \\
&= \bar{A}_k \left( I - \hat{P}_{k|k} G_{k11} \right) + \left( \bar{A}_k \hat{P}_{k|k} G_{k12} - \bar{B}_k \right) \times U_k G_{k12}^T \left( I - \hat{P}_{k|k} G_{k11} \right) \\
&= \left( \bar{A}_k - \bar{B}_k U_k G_{k12}^T \right) \left( I - \hat{P}_{k|k} G_{k11} \right) = {}^{\triangle}A_k(0).
\end{aligned}
\tag{A31}
$$

Thus, $A_{pk}$ can be simplified to $\hat{A}_k \left( I + P_{k|k-1} \bar{C}_k^T R_k^{-1} \bar{C}_k^T \right)^{-1}$. Because of $A_{fk} = \left( I + P_{k+1|k} \bar{C}_{k+1}^T R_{k+1}^{-1} \bar{C}_{k+1} \right){}^{\triangle}A_k(0)$, (A32) can be obtained.

$$
A_{fk} = \left( I + P_{k+1|k} \bar{C}_{k+1}^T R_{k+1}^{-1} \bar{C}_{k+1} \right)^{-1} A_{pk} \left( I + P_{k|k-1} \bar{C}_k^T R_k^{-1} \bar{C}_k \right)
\tag{A32}
$$

It can be known from Theorem 1 that $P_{k|k-1}$ converges to a constant matrix under certain conditions. When the nominal system matrix is assumed to be time-invariant, this convergence means that $\lim_{k\to\infty} \left( P_{k+1|k} \bar{C}_{k+1}^T R_{k+1}^{-1} \bar{C}_{k+1} - P_{k|k-1} \bar{C}_k^T R_k^{-1} \bar{C}_k \right) = 0$. It can be inferred from the above equation that as k increases, the set of eigenvalues of $A_{fk}$ converges to the set of eigenvalues of $A_{pk}$, and the latter converges to a stable constant matrix. Assuming the conditions in Theorem 1 are satisfied, the robust state estimator converges to a linear time-invariant stable system. Define $X_k, \hat{X}_{k|k}, \tilde{X}_{k|k}$ as $X_k = [I + \Gamma_k(0)]\bar{x}_k$, $\hat{X}_{k|k} = [I + \Gamma_k(0)]\hat{\bar{x}}_k$, $\tilde{X}_{k|k} = X_k - \hat{X}_{k|k}$. Then, (A33) can be obtained directly from (A26) and the derivation process of (A30) and (A31).

$$
\begin{bmatrix} \tilde{X}_{k+1|k+1} \\ \hat{X}_{k+1|k+1} \end{bmatrix} = \tilde{A}_k(\varepsilon_k, \varepsilon_{k+1}) \begin{bmatrix} \tilde{X}_{k|k} \\ \hat{X}_{k|k} \end{bmatrix} + \tilde{B}_k(\varepsilon_k, \varepsilon_{k+1}) \begin{bmatrix} w_k \\ v_{k+1} \end{bmatrix},
\tag{A33}
$$

where

$$
\tilde{A}_k(\varepsilon_k, \varepsilon_{k+1}) = \begin{bmatrix} \tilde{A}_{k11} & \tilde{A}_{k12} \\ \tilde{A}_{k21} & \tilde{A}_{k22} \end{bmatrix}, \tilde{B}_k(\varepsilon_k, \varepsilon_{k+1}) = \begin{bmatrix} \tilde{B}_{k11} & \tilde{B}_{k12} \\ \tilde{B}_{k21} & \tilde{B}_{k22} \end{bmatrix},
\tag{A34}
$$

and

$$
\Lambda_k(\varepsilon_k) = (I + \Gamma_{k+1}(0)) \begin{bmatrix} A_k(\varepsilon_k) & 0_{n\times(d-1)m} & 0_{n\times m} \\ C_k(\varepsilon_k) - C_{k+1}(\varepsilon_{k+1}) A_k(\varepsilon_k) & 0_{n\times(d-1)m} & 0_{n\times m} \\ 0_{(d-1)m\times n} & I_{(d-1)m} & 0_{(d-1)m\times n} \end{bmatrix}
\tag{A35}
$$

$$
\begin{cases}
\tilde{A}_{k11} = \left( I + \Gamma_{k+1}(0) - B_{fk}\bar{C}_{k+1}(\varepsilon_{k+1}) \right)(I + \Gamma_{k+1}(0))^{-1}\Lambda_k(\varepsilon_k) \\
\tilde{A}_{k21} = B_{fk}\bar{C}_{k+1}(\varepsilon_{k+1})(I + \Gamma_{k+1}(0))^{-1}\Lambda_k(\varepsilon_k) \\
\tilde{A}_{k22} = \tilde{A}_{k21} + A_{pk}, \tilde{A}_{k12} = \tilde{A}_{k11} - A_{pk} \\
\tilde{B}_{k11} = \left( (I + \Gamma_{k+1}(0)) - B_{fk}\bar{C}_{k+1}(\varepsilon_{k+1}) \right)\bar{B}_k(\varepsilon_k) \\
\tilde{B}_{k21} = B_{fk}\bar{C}_{k+1}(\varepsilon_{k+1})\bar{B}_k(\varepsilon_k) \\
\tilde{B}_{k12} = -B_{fk}, \tilde{B}_{k22} = B_{fk}
\end{cases}
\tag{A36}
$$

Based on the above relationship and the stability of matrix $\bar{A}_k(\varepsilon_k)$, the estimation error covariance matrix of the robust state estimator can be obtained with bounded and asymptotically unbiased conditions. Note that (A33) is similar to (16) in [40], and it can be proved in the same way, which is omitted here.

Theorem 2 is proved.

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
