# Peer review of "Robust State Estimation for Uncertain Discrete Linear Systems with Delayed Measurements"

_mathematics, doi:10.3390/math10091365_

Round 1

Reviewer 1 Report

This paper presents robust state estimation for discrete-time linear systems with parametric uncertainties and delay measurements. The design method combining Kalman filter with RLS framework and state augmentation. Stability analysis of the proposed estimator and numerical examples are provided. Please find the following comments:
1. The proposed method aiming to cope simultaneous problems of parametric uncertainties and delay measurements is interesting. However, the measurement delay here is constant and known which makes the delay problem is less challenging.  Some assumption required in the design including the type of the measurement delay should be given.
2.  It is quite confusing when the experimental verification is mentioned in Section 4, but no experimental setup is described.
3.  It has been mentioned that the authors work on the photoelectric tracking systems. Does the simulated model represent the photoelectric tracking system?
4.  Line 189, what theorem?? 

Author Response

Concern # 1: The proposed method aiming to cope simultaneous problems of parametric uncertainties and delay measurements is interesting. However, the measurement delay here is constant and known which makes the delay problem is less challenging. Some assumption required in the design including the type of the measurement delay should be given.

Author response: Thanks for your constructive comments. Some detailed information of measurement delay has been given in the second paragraph of INTRODUCTION. The commonly used detectors in our research filed are photoelectric detectors with constant measurement delay which depends on the exposure time and image processing time. Thus, we focus only on the constant measurement delay problem in this article. The measurement acquired instant is also set to be 3 frames after the real state like practical photoelectric tracking system. In our simulations, the used delay time is given in Line 197, 223 as 3 frames and their type are set as constant.

Concern # 2: It is quite confusing when the experimental verification is mentioned in Section 4, but no experimental setup is described.

Author response: Thanks for your detailed criticism. We neglected to modify this place from our article template. Words “and experimental verification” had been deleted.

Concern # 3: It has been mentioned that the authors work on the photoelectric tracking systems. Does the simulated model represent the photoelectric tracking system?

Author response: The simulated model can’t be seemed as representing the whole photoelectric tracking system. We focused on the acquisition and tracking problems in this paper rather than the control method of the photoelectric tracking system which is another broad field and worth being fully consideration. Precisely, the simulated model is refined from a part of the photoelectric tracking system. Main issues in this paper, uncertainty in state transition equation and the delay in measuring, are truly exist due to target’s uncertain motion model and the detector’s signal delay. Specifically, the uncertain motion model is caused by target’s maneuvering or environmental factors e.g., air drag, atmosphere turbulence and so on, which are hard to model and predict precisely, and thus the robust estimator is needed. As for the detector signal delay, it is composed of the exposure time, the sampling time, and the image processing time, which is invariable in the photoelectric tracking system and needed to be compensated.

Concern # 4: Line 189, what theorem??

Author response: Thanks for your detailed criticism. The composing problem has been corrected.

Reviewer 2 Report

The authors proposed an algorithm for discrete linear systems with delayed measurements. The paper organised on 5 chapters starting with an extended introduction followed by analysis and explanation of the problem statement and the design of robust state estimator. In the third chapter, the authors present the recursive procedure and the stability conditions. The work from previous chapter is sustained in the fourth chapter by numerical simulations to verify the proposed method. This is a well written paper of an improvement of Kalman filter that is very popular in the field and well-studied for its performances.  If it is not the target of this article, in the future, I propose that the authors move from simulative environment of the processing of data to the real environment.

Author Response

Author response: Thank you for the constructive comments. We agree with the reviewer that the proposed estimator needs to be discussed in more detail and supported with experimental results. We are constructing an experimental platform with a double reflector. We will apply the proposed estimator to the actual photoelectric tracking system and verify its effectiveness in real environment.

Reviewer 3 Report

In the submitted manuscript, a robust estimation method for discrete-time linear systems based on the combination of Kalman filter regularized least-squares framework and state augmentation is proposed. Constant measurement delay and random model parametric uncertainties are supposed. Numerical examples are given to the reader to justify the theoretical results.

The manuscript is well written, but the reviewer has one major issue and some minor comments.

Comments:

1) The most crucial criticism: The reviewer wonders why no previous author’s work on the topic is referenced. For instance, in DOI: 10.1049/cth2.12153, a very close problem via similar tools has been solved by the authors. What is the difference between the submitted research and the referenced article? Is the proposed method better? Can it be proved at least by simulations? For instance, the shape of Figure 6 looks almost like the shape of Figure 3 from the referenced article… This issue must be elaborated in more detail for the reader.

Besides, some important results by other researchers are ignored, see, e.g., DOI:  10.1007/s11768-014-4072-4.

2) In many places, there is “Theorem ??”. I.e., the theorem number is missing. See, e.g., lines 189, 228, 282.

The reviewer means that issue no. 1 should be thoroughly addressed in the revised version.

Author Response

Concern #1: The reviewer wonders why no previous author’s work on the topic is referenced. For instance, in DOI: 10.1049/cth2.12153, a very close problem via similar tools has been solved by the authors. What is the difference between the submitted research and the referenced article? Is the proposed method better? Can it be proved at least by simulations? For instance, the shape of Figure 6 looks almost like the shape of Figure 3 from the referenced article… This issue must be elaborated in more detail for the reader.

Besides, some important results by other researchers are ignored, see, e.g., DOI:  10.1007/s11768-014-4072-4.

Author response: Thanks for your constructive comments. The tool used in article DOI: 10.1049/cth2.12153 is truly similarly to this article in some extent. The author who wrote the original draft neglected this previous work. We are glad to refer it in our article now and you can see the added reference in Line 49-51 in our new manuscript. As for the difference between these two articles, they are presupposed to solve different problems. The problem in DOI: 10.1049/cth2.12153 is the d-step preceding historical state will affect the change of current state. While the refined problem in this article is measurement delay, which has no influence in target’s state transition. The validity of our estimator has been demonstrated in the simulation part while it’s hard to be compared with the method in DOI: 10.1049/cth2.12153, because they are designed under different assumptions. As for the shape of Figure 6 looks similarly to the shape of Figure 3 in DOI: 10.1049/cth2.12153. They are analyzed through same statistical approach, but they are different. We are also glad to refer the valuable research result DOI: 10.1007/s11768-014-4072-4, you can see it in Line 85 in our new manuscript.

Concern #2:  In many places, there is “Theorem ??”. I.e., the theorem number is missing. See, e.g., lines 189, 228, 282.

Author response: Thanks for your detailed criticism. The composing problem has all been corrected.

Round 2

Reviewer 1 Report

This paper has been revised accordingly. We have no further comment. Thank you.

Author Response

Thanks for your constructive comments and detailed criticisms

Reviewer 3 Report

The authors have reflected on all the reviewer’s comments. However, two minor issues might be addressed (one of them is related to the preceding comments):

1) P. 2, l. 49-51: The reviewer appreciates that ref. [18] has been added. However, it would be useful to provide the reader with the information given in the “Response to reviewers”.

“While the refined problem in this article is measurement delay, which has no influence in target’s state transition. The validity of our estimator has been demonstrated in the simulation part while it’s hard to be compared with the method in DOI: 10.1049/cth2.12153, because they are designed under different assumptions.”

Please, rephrase these sentences later in the Introduction or in Section 4 to give the reader full information.

2) P. 2, l. 74: A missing blank space (“estimate[22,23].”).

After fixing these two minor issues, the manuscript can be accepted for publication.

Author Response

Reviewer#3, Concern # 1: P. 2, l. 49-51: The reviewer appreciates that ref. [18] has been added. However, it would be useful to provide the reader with the information given in the “Response to reviewers”.

While the refined problem in this article is measurement delay, which has no influence in target’s state transition. The validity of our estimator has been demonstrated in the simulation part while it’s hard to be compared with the method in DOI: 10.1049/cth2.12153, because they are designed under different assumptions.”

Please, rephrase these sentences later in the Introduction or in Section 4 to give the reader full information.

After fixing these two minor issues, the manuscript can be accepted for publication.

Author response: Thanks for your constructive comments. Corresponding sentence in Introdunction has been substituted to “One estimator based on augmented state vector is proposed in [18] to deal with the problem that the change of current state could be affected by d-step preceding state. However, it could not be adopted to solve the problem in this paper that detectors have measurement delay and preceding state has no influence in target's state transition.”

Reviewer#2, Concern # 2: P. 2, l. 74: A missing blank space (“estimate[22,23].”).

Author response: Thanks for your detailed criticism, the missing blank space has been added.